# Microevolution, reinfection and highly complex genomic diversity in patients with sequential isolates of *Mycobacterium abscessus*

Sergio Buenestado-Serrano [1,2,3], Miguel Martínez-Lirola[4], Marta Herranz-Martín[1,2], Jaime Esteban [5,6], Antonio Broncano-Lavado[5], Andrea Molero-Salinas[1,2], Amadeo Sanz-Pérez[1,2], Jesús Blázquez[7], Alba Ruedas-López [8], Carlos Toro[9], Paula López-Roa[8], Diego Domingo[10], Ester Zamarrón [11], María Jesús Ruiz Serrano[1,2,12], Patricia Muñoz[1,2,12,13], Laura Pérez-Lago [1,2,14] ✉ & Darío García de Viedma [1,2,12,14] ✉

*Mycobacterium abscessus* is an opportunistic, extensively drug-resistant non-tuberculous mycobacterium. Few genomic studies consider its diversity in persistent infections. Our aim was to characterize microevolution/reinfection events in persistent infections. Fifty-three sequential isolates from 14 patients were sequenced to determine SNV-based distances, assign resistance mutations and characterize plasmids. Genomic analysis revealed 12 persistent cases (0-13 differential SNVs), one reinfection (15,956 SNVs) and one very complex case (23 sequential isolates over 192 months), in which a first period of persistence (58 months) involving the same genotype 1 was followed by identification of a genotype 2 (76 SNVs) in 6 additional alternating isolates; additionally, ten transient genotypes (88-243 SNVs) were found. A macrolide resistance mutation was identified from the second isolate. Despite high diversity, the genotypes shared a common phylogenetic ancestor and some coexisted in the same specimens. Genomic analysis is required to access the true intra-patient complexity behind persistent infections involving *M. abscessus*.

Mycobacterium abscessus, a global, non-tuberculous, rapidly-growing mycobacterium, is considered an opportunistic pathogen. Given its emerging nature and its drug-resistant profile, *M. abscessus* is a clinically relevant mycobacterium, especially in patients with cystic fibrosis (CF).

Several recent studies have applied genomic analysis to identify whether interpatient transmission of *M. abscessus* between CF cases is a possibility, as an alternative explanation to the general assumption of environmental exposure as the origin of infection. However, this question remains unsolved, as some studies pointed to frequent clustering of CF patients[1], whereas others found no differences in the frequency of inclusion in clusters of CF and non-CF patients[2,3]. Furthermore, even though clusters of patients seen at the same hospital have been found, epidemiological links between the patients are frequently not found[4]. Meanwhile, clustering of patients from different institutions and even living in different countries has also been detected[5].

Some studies[6] have applied genomic analysis in order to determine the intrapatient diversity of *M. abscessus*. Coinfections

with more than one strain[7], and even more than one subspecies[8], have been described[1,9,10]. Most analyses involving more than one longitudinal isolate per patient have found that the number of SNVs between them is low, and always below the genomic threshold defined to consider relatedness in interpatient genomic analysis, namely 25–30 SNPs[3,4,10,11]. Differentiating between persistence and reinfection is crucial in *M. abscessus*, as it can point either to therapeutic adjustments or to search for new causes for a re-exposure, respectively.

Long term *M. abscessus* infections could likely lead to intrapatient acquisition of diversity, including potential resistance emergence. This diversity might be distributed heterogeneously, leading to the distribution of different evolved variants in different lung sites. This clonal complexity might not be identified in a standard, single sputum, analysis and therefore more exhaustive longitudinal studies are required to analyze in detail this hypothesis.

The focus of our study is the application of genomic characterization to provide a more complete description of persistence, microevolution and reinfections in a group of patients with sequential *M. abscessus* isolates. Our findings indicate that clonal complexity over prolonged infections surpasses common assumptions, raising concerns about the criteria typically considered for determining epidemiological relationships in this mycobacterium.

## Results

### Genomic analysis of sequential isolates

Fourteen patients with two or more sequential *M. abscessus* isolates were included in the study, resulting in 53 isolates from the period 2007–2023. The time between first and last isolates for each patient ranged from 2 months to 16 years, and the number of isolates per patient was 2–23 (Table 1). Five patients had cystic fibrosis, 4 bronchiectasis, three other diseases (obstructive sleep apnea syndrome, respiratory infection and HIV pneumonia), and no information was available on the underlying disease of the other two patients. Genomic analysis identified the subspecies involved: three patients were infected with *M. abscessus subsp. massiliense* (Table 1, Fig. 1a) and the remaining eleven with *M. abscessus subsp. abscessus* (five of them involving the dominant circulating clone 1 (DCC1); Table 1; Fig. 1b).

Genomic distances, measured as the number of single nucleotide variants (SNVs), between the strains infecting the different patients (Supplementary Information, Supplementary Table 1) indicated the involvement of different strains. When determining the genomic distances for the intrapatient sequential isolates, in twelve patients (patients 1–12, 2–46 months apart), they ranged between 0 and 13 SNVs (Table 1; Supplementary Data 1), which were therefore considered as persistence infections with/without microevolution (four were CF patients). Nine of these corresponded to *M. abscessus subsp. abscessus*, and the remaining three to *M. abscessus subsp. massiliense*. There was no direct relationship between number of SNVs and time between isolates; we observed sequential isolates 8–12 months apart with 0 SNVs between them, whereas 7 SNVs were identified between two of the closest isolates (4 months, patient 11). Regarding extrachromosomal elements, a plasmid was identified only in Patient 3 (*Mycobacterium abscessus subsp. abscessus strain* GD69A, plasmid pGD69A-1).

Another patient (patient 13) with much greater genomic distance (15,956 SNVs) between the two sequential isolates (76 months apart) was more consistent with a reinfection involving two different strains (Fig. 1b; Table 1). This corresponded to *M. abscessus subsp. abscessus* in a patient with diffuse bronchiectasis diagnosed with Lady Windermere syndrome.

The remaining case (patient 14) had the highest number of isolates over the longest observation period (23 isolates over 16 years) and deserves a separate in-depth analysis.

### Clinical-therapeutic features from Patient 14

Patient 14, a 15-year-old female (at the first isolation of *M. abscessus*) had severe homozygous F508 del cystic fibrosis, diagnosed at 2 years of age with pulmonary, digestive, and hepatic manifestations, and exocrine and endocrine pancreatic insufficiency. For most of the period of persistent infection with *M. abscessus* (between diagnosis and 2019, over 13 years), she received inhaled colistin and clarithromycin. This treatment was reinforced at different times during the persistent infection (Fig. 2) with other antibiotics (amikacin, cefoxitin, levofloxacin, trimethoprim-sulfamethoxazole, aztreonam, tobramycin, azithromycin), especially in the period 2009–2020, between mid-2015 and 2019, and more intensively in the last four years by combining several antibiotics and CFTR modulators (Fig. 2; Supplementary Information, Supplementary notes). Some of the therapeutic changes were made after new isolates of *M. abscessus* were observed indicating persistence of infection (Fig. 2).

### Detailed genomic analysis from sequential cultures from patient 14

Genomic analysis (Fig. 2) indicated that patient 14 had a period of persistence (seven sequential isolates over 58 months) involving closely related isolates (Genotype 1; 0–12 SNVs between isolates). In the September 2013 isolate (month 77), 19 months after the last Genotype 1 isolate, the number of SNVs detected (76) was higher, thus defining a new genotype (Genotype 2). After that, Genotype 1 was not detected in subsequent isolates, while Genotype 2 continued to be detected in another five subsequent isolates (differing by 0–9 SNVs), taken in months 101, 103, 120, 140 and 189 (Fig. 2). In addition to the intermittent detection of Genotype 2, ten new genotypes (Genotypes 3–12) were detected in months 96, 108, 117, 129, 130 (2), 134, 147, 155 and 192, with an increased number of SNVs between them (average 289 SNVs; Supplementary Information, Supplementary Table 2 and Supplementary Data 1) and with respect to genotypes 1 and 2 (from 164/88 to 319/243 pairwise distances, respectively); each genotype was detected in only one of the sequential specimens. It was noted that one of the genotypes (Genotype 8) was identified in a bronchoalveolar lavage (BAL) specimen, while a different one (Genotype 7) was identified in a sputum specimen taken nine days apart (pairwise distance between Genotypes 7 and 8: 362 SNVs).

Because of the high number of different genotypes (all corresponding to DCC1) detected in this case, we repeated the analysis on all specimens showing differences; after re-extraction and re-sequencing of the samples, all different genotypes were confirmed. We used STR-based host genetic analysis to confirm that all specimens belonged to the same patient and to rule out possible sample mislabeling or mishandling in the laboratory.

Of note, the patient went through a period of marked exacerbation of symptoms with hemoptysis, which led to hospitalization in an ICU in January 2018. Coincidentally, during that period, within a relatively short period of 28 days, three different genotypes (Genotypes 6, 7 and 8) were identified.

### Phenotypic features from genomic characterization in Patient 14

Colony phenotypes were determined in 14 of the isolates, associated with genotypes 1 (4), 2 (4), 3, 4, 5, 6, 9 and 11, and all corresponded to the rough phenotype (Fig. 2). All harbored indels in the GPL gene cluster (a 1-bp insertion in the methyltransferase MtfD gene (*rmt3*) in isolate 1, and an additional 20-bp insertion in the peptide synthetase NRP (*mps*2) in all subsequent isolates).

Biofilm production in the sequential isolates varied widely (Fig. 3; Source Data, Stepanovich_Results.xlsx), although the range of production in the majority was between 2 and 3 (moderate). No correlation was found between mean biofilm formation and chronology of infection as represented by the sequential isolates (Pearson *p*-value = 0.1309). Nor were there differences in mean production between the

**Table 1 | Information on patients and isolates in the study, including their underlying diseases, sampling date and time between first and last isolate**

| Patient | #Isolate | Disease | Sampling Date | Time-Lapse | Subspecies | SNVs | Interpretation | ST | DCC | Accession Number |
|---|---|---|---|---|---|---|---|---|---|---|
| Patient 1 | 1-1 | No Data | 2013-04-19 | 3 months | *abscessus* | 0 | Persistence | 47 | – | ERS14874751 |
| | 1-2 | | 2013-07-24 | | | | | 47 | – | ERS14874752 |
| Patient 2 | 2-1 | Paracardiac bronchiectasis (Chronic bronchitis) | 2008-05 | 8 months | *abscessus* | 0 | Persistence | – | 1 | ERS14874753 |
| | 2-2 | | 2009-01 | | | | | 5 | 1 | ERS14874754 |
| Patient 3 | 3-1 | No Data | 2018-05-22 | 8 months | *massiliense* | 0 | Persistence | 46 | – | ERS14874755 |
| | 3-2 | | 2019-01-03 | | | | | 46 | – | ERS14874756 |
| Patient 4 | 4-1 | Cystic fibrosis | 2015-11 | 1 year | *abscessus* | 0 | Persistence | 28 | – | ERS14874757 |
| | 4-2 | | 2016-11 | | | | | 28 | – | ERS14874758 |
| Patient 5 | 5-1 | Diffuse bronchiectasis | 2015-06 | 2 years | *abscessus* | 0-1 | Persistence + Microevolution | – | – | ERS14874759 |
| | 5-2 | | 2016-11 | | | | | – | – | ERS14874760 |
| | 5-3 | | 2017-06 | | | | | – | – | ERS14874761 |
| Patient 6 | 6-1 | Cystic fibrosis | 2019-08-14 | 2 months | *massiliense* | 1 | Persistence + Microevolution | 33 | 3 | ERS14874762 |
| | 6-2 | | 2019-10-10 | | | | | 33 | 3 | ERS14874763 |
| Patient 7 | 7-1 | Cystic bronchiectasis | 2012-05 | 9 months | *abscessus* | 1 | Persistence + Microevolution | – | – | ERS14874764 |
| | 7-2 | | 2013-02 | | | | | – | – | ERS14874765 |
| Patient 8 | 8-1 | Obstructive Sleep Apnea Syndrome | 2008-12 | 2 months | *abscessus* | 2 | Persistence + Microevolution | – | 1 | ERS14874766 |
| | 8-2 | | 2009-02 | | | | | – | 1 | ERS14874767 |
| Patient 9 | 9-1 | Cystic fibrosis | 2018-05-18 | 1 year & 8 months | *massiliense* | 2 | Persistence + Microevolution | 33 | 3 | ERS14874768 |
| | 9-2 | | 2020-01-29 | | | | | 33 | 3 | ERS14874769 |
| Patient 10 | 10-1 | Cystic fibrosis | 2018-08-08 | 11 months | *abscessus* | 3-4 | Persistence + Microevolution | 40 | – | ERS14874770 |
| | 10-2 | | 2018-12-14 | | | | | 40 | – | ERS14874771 |
| | 10-3 | | 2019-07-19 | | | | | 40 | – | ERS14874772 |
| Patient 11 | 11-1 | Respiratory Infection | 2010-10-07 | 4 months | *abscessus* | 7 | Persistence + Microevolution | 77 | – | ERS14874773 |
| | 11-2 | | 2011-02-09 | | | | | 77 | – | ERS14874774 |
| Patient 12 | 12-1 | HIV pneumonia | 2008-12-09 | 3 years & 10 months | *abscessus* | 3-13 | Persistence + Microevolution | 5 | 1 | ERS14874775 |
| | 12-2 | | 2011-01-13 | | | | | 5 | 1 | ERS14874776 |
| | 12-3 | | 2011-05-09 | | | | | 5 | 1 | ERS14874777 |
| | 12-4 | | 2012-10-11 | | | | | 5 | 1 | ERS14874778 |
| Patient 13 | 13-1 | Diffuse bronchiectasis (Lady Windermere Syndrome) | 2014-03 | 6 years & 4 months | *abscessus* | 15,956 | Reinfection | 155 | – | ERS14874779 |
| | 13-2 | | 2020-07 | | | | | 5 | 1 | ERS14874780 |
| Patient 14 | 14-1 | Cystic fibrosis | 2007-04-01 | 16 years | *abscessus* | 0-475 | Persistence + Microevolution | 5 | 1 | ERS14874781 |
| | 14-2 | | 2009-02-11 | | | | | 5 | 1 | ERS14874782 |
| | 14-3 | | 2009-02-12 | | | | | 5 | 1 | ERS14874783 |
| | 14-4 | | 2010-03-11 | | | | | 5 | 1 | ERS14874784 |
| | 14-5 | | 2010-09-24 | | | | | 5 | 1 | ERS14874785 |
| | 14-6 | | 2011-10-26 | | | | | 5 | 1 | ERS14874786 |
| | 14-7 | | 2012-02-16 | | | | | – | 1 | ERS14874787 |
| | 14-8 | | 2013-09-19 | | | | | 5 | 1 | ERS14874788 |
| | 14-9 | | 2015-04-16 | | | | | 5 | 1 | ERS14874789 |
| | 14-10 | | 2015-09-09 | | | | | 5 | 1 | ERS14874790 |
| | 14-11 | | 2015-11-09 | | | | | – | 1 | ERS14874791 |
| | 14-12 | | 2016-04-22 | | | | | 5 | 1 | ERS14874792 |
| | 14-13 | | 2017-01-16 | | | | | 5 | 1 | ERS14874793 |
| | 14-14 | | 2017-04-10 | | | | | 5 | 1 | ERS14874794 |
| | 14-15 | | 2018-01-10 | | | | | – | 1 | ERS14874795 |
| | 14-16 | | 2018-01-29 | | | | | – | 1 | ERS14874796 |
| | 14-17 | | 2018-02-07 | | | | | 5 | 1 | ERS14874797 |
| | 14-18 | | 2018-06-04 | | | | | 5 | 1 | ERS14874798 |
| | 14-19 | | 2018-12-17 | | | | | 5 | 1 | ERS14874799 |
| | 14-20 | | 2019-07-22 | | | | | 5 | 1 | ERS14874800 |
| | 14-21 | | 2020-03-16 | | | | | 5 | 1 | ERS14874801 |
| | 14-22 | | 2023-01-05 | | | | | 5 | 1 | ERS15972583 |
| | 14-23 | | 2023-04-13 | | | | | – | 1 | ERS15972584 |

Species assignment based on genomic analysis, SNV differences between isolates from the same patient and interpretation of results based on established SNV thresholds are included. Additionally, the sequence type (ST) obtained through multilocus sequence typing (MLST) analysis and the dominant circulating clone (DCC) assigned through phylogeny are included. Cells with '-' are either due to a failure to assign ST (below 100% match) or to not belonging to any DCC. Accession numbers for the sequences are included.

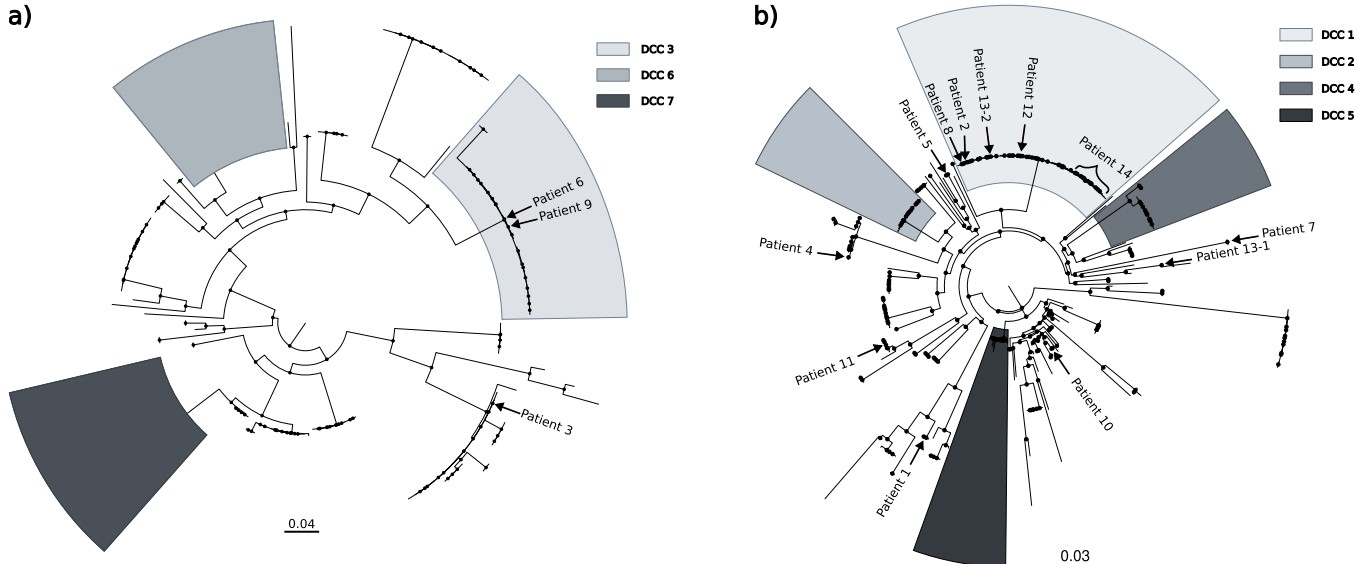

**Fig. 1 | Phylogenetic analysis of *M. abscessus* subspecies and clonal diversity.** Phylogenetic tree for the (**a**) *M. abscessus subsp masiliense* and (**b**) *M. abscessus subsp abscessus*, sequences from our study (labeled as Patient 1–14) together with

150 and 252 sequences, respectively, representative of different DCCs. Scale corresponds to nucleotide divergence (SNPs/site).

two persistent genotypes (Genotype 1 and 2; Tukey multiple mean comparison *p*-value 0.8277). Nevertheless, the two persistent genotypes 1 and 2 produced more biofilm than isolates with the non-persistent intermittent genotypes (Welch's t-test *p*-values of 0.0128 and 0.0110 respectively).

## Phylogenetic analysis from genotypes identified in Patient 14

A phylogenetic analysis was performed by integrating all genotypes found in patient 14 with 362 additional *M. abscessus* sequences[12], assuring i) an enrichment in sequences from the same dominant circulant clone (DCC 1) which was involved in patient 14 infection, and ii) the inclusion of all 13 DCC1 sequences available from Spain in SRA. The phylogeny showed closer relationships among genotypes from patient 14, sharing the same deep private branch and ancestor (Fig. 4). A Bayesian tree was also obtained (Fig. 5) and it showed how all genotypes evolved from genotype 1 and then diversified along two evolving branches, one including genotypes 2, 4, 5, 6, 8, 10 and 11, and the other branch including genotypes 3, 7, 9 and 12.

## Analysis of coexistence of genotypes in Patient 14

Following the unexpectedly high number of different genotypes identified in patient 14 (some of them identified close in time to each other, Figs. 2 and 5), we evaluated whether some of them were coexisting in some of the specimens analyzed. Genotypic coexistence in the same specimen should give rise to a number of heterozygous (HZ) calls in the sequencing reads with values above the mean. HZ calls are often removed by bioinformatic pipelines to minimize sequencing errors and are not usually taken into account in genomic analysis. When we focused on HZ calls that were not expected to be due to sequencing errors (≥20× of the alternative allele with allele frequency between 0.2 and 0.7), the number of positions with HZ calls observed in some of the six sequential cultures in which we had identified genotype 2 (now reclassified as 2, 2.1, 2.2, 2.3 2.4 and 2.5, following the sampling chronology) was higher than expected (reaching 187–390 HZs in genotypes 2.2, 2.3, 2.4 and 2.5; Table 2). We speculate that this increased number of HZ calls could be due to the coexistence of genotypes.

First, we focused on the Patient 14 genotypes, to check whether any of the alleles involved in the HZ calls identified in each of the

genotypes, corresponded to the alleles fixed as SNVs in the remaining genotypes (Table 2). We observed that 1-75 of the HZ calls in certain genotypes corresponded to SNVs identified in other genotypes: SNVs from genotypes 2, 5, 8 and 10 where those responsible for the HZ calls identified in other Patient 14 genotypes (Table 2).

As a second step, we assessed whether other genotypes related to those identified in our analysis, but not identified as single genotypes in the specimens tested, would explain some proportion of the HZ calls found in the genotypes under analysis. For this second analysis we selected SNVs common to the intermediate nodes in the tree (nodes 1–5, Fig. 5), rather than those specific to the Patient 14 genotypes. SNVs corresponding to the intermediate nodes explained between 3–186 of the HZ calls in the genotypes under study (Table 2). Our observations suggest that other evolutionary genotypes, not identified as isolated/single genotypes in the specimens studied in our analysis, could also co-exist.

In summary, the total SNVs contributed by the genotypes and intermediate nodes explained from 1.64 to 73.44% of the heterozygous identified in our study. For some specific genotypes (2.1, 2.4 and 2.5 they explained a high percentage of the HZs calls identified: 65.63, 73.44 and 66.92, respectively reinforcing the hypothesis of the simultaneous coexistence of several genotypes in some patient specimens. On the other hand, in genotypes 2.2 and 2.3, which also had an unusually high number of HZ calls, only 33.48% and 16.04%, respectively of these HZs, could be explained, suggesting coexistence with other genotypes not represented in the samples handled in the study.

## Resistance mutations in Patient 14

In terms of resistance mutations, we checked for the presence of mutations at A2270, A2271, G2281 and A2293 (or A2058, A2059, G2069 and A2082 using *E. coli* numbering) in the *rrl* gene[2], responsible for acquired resistance to clarithromycin and macrolides. We identified A2270C in isolated culture 2 at an allele frequency of 65%, which was found to be fixed (100% frequency) in isolate 3 of Genotype 1. All subsequent genotypes since then have harbored this fixed mutation. We also searched for mutations T1372, A1374, C1375, G1458, C1463, T1465 (or T1406, A1408, C1409, G1491, C1496, T1498 with *E. coli* numbering) in the *rrs* gene (confers resistance to aminoglycosides); no mutations were detected.

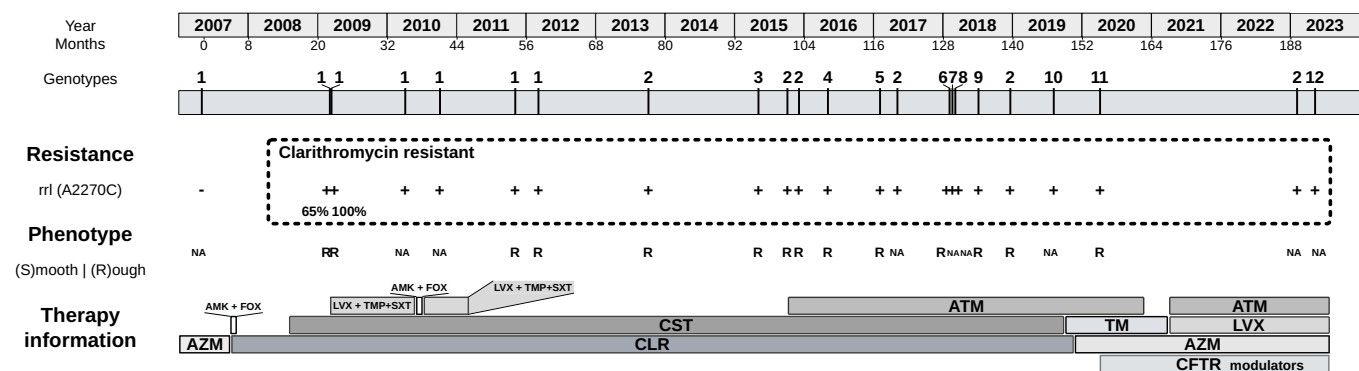

**Fig. 2 | Timeline for isolates collected from patient 14, showing genotype assignment based on genomic distance.** Resistance results, phenotype and therapeutic information, including treatment received are also shown. Month 0 begins at the time of collection of the first isolate available for our study. AMK amikacin, AZM azithromycin, ATM aztreonam, CFTR modulators, CLR clarithromycin, CST colistin, FOX cefoxitin, LVX levofloxacin, TMP-SXT trimethoprim-sulfamethoxazole, TM tobramycin.

## Analysis of hypermutator genotypes in Patient 14

To evaluate a potential involvement for some hypermutator phenotype in Patient 14 genotypes, we checked the presence of SNVs or rearrangements in 65 genes (Supplementary Information, Supplementary Table 3) potentially associated with hypermutation (coding for peroxidases, catalases, genes involved in DNA repair, etc). A non-synonymous change (Val617Met in MAB_3516c, encoding the uvrD-like helicase protein) was identified in all genotypes except in genotype 1 and an indel (Thr143fs in MAB_3543c, encoding the RNA polymerase sigma-E factor) was also identified in all genotypes and in two representatives of Genotype 1 (isolates 2 and 7). In addition, different non-synonymous variants (7) were identified; three in genotype 3 (Pro67Ser in MAB_1160, Asn84Ser in MAB_2712c and Glu145Gly in MAB_3909); one in genotypes 6 and 10 (Arg66His in the MAB_1349 gene) three in genotypes 5, 8 and 11 each (Val217Ala in MAB_3480, Thr109Ala in MAB_3543c and Thr39Ala in MAB_4408c). Of note, in the hybrid assembly obtained as a reference for all the analysis in Patient 14 (see Methods), two genes from the HNH endonuclease family were involved in gene rearrangement events (PEG.2664 and PEG.5299, with 100% of coverage and identity with Mycobacteroides genus HNH endonuclease (WP_052524998.1) and with MAB HNH endonuclease (WP_062923687.1), respectively) which are not present in the ATCC reference strain for *M. abscesssus ssp abscessus*.

## Plasmid analysis in Patient 14

Plasmid analysis on sequential isolates of patient 14 indicated the presence of 2 plasmids (Plasmid 1: 137,682 bp and Plasmid 2: 12,867 bp) in all of them except for specimens 18 and 23 (corresponding to Genotype 9 and 12), which showed an incomplete plasmid 1. A BLAST search against the entire NCBI database yielded only a low-similarity match (30% of query coverage, 41,570 bp of total plasmid length aligned with 99.873% identity) to *Mycobacterium abscessus* JCM 30620 plasmid pJCM30620_1 (AP022622.1), while no identification was made using PlasmidID. Plasmid 2 was identified (99.33% and 99.82% identity) by both BLAST and PlasmidID as *Mycobacterium avium subsp. hominissuis* strain OCU901_s_S2_2s plasmid pS2c (NZ_CP076861.1).

## Discussion

Several studies have applied genomic analysis to an intrapatient analysis of *M. abscessus* to describe within-host genomic diversity[1,6,9,10]. In twelve of our fourteen patients in our study, the same genotype was identified, and the diversity acquired was modest (0-13 SNVs), below the genomic threshold defined to consider relatedness in interpatient genomic analysis, namely 25-30 SNPs[3,4,10,11], comparable to other studies[4,8,13]. In only one of our cases did we find a much greater

genomic distance between the sequential genotypes, compatible with reinfection involving a different strain.

In another one of our patients, we had the opportunity to study the *M. abscessus* infection dynamics along a really prolonged infection, namely 16 years. This case corresponded to a CF patient, in whom *M. abscessus* infection was first detected when she was a child, two features that made her therapeutical management a big challenge. These clinical circumstances also forced to deviating from established guidelines, which recommends avoiding macrolide monotherapy (for infections with macrolide susceptible strains) and promote combination therapy with macrolides plus proven companion intravenous drugs. The clinicians difficulties to follow the guidelines, at certain moments, were the result of priorizing patient stability, mitigate toxicity, assure adherence and tolerance in a long term scope, with the additional complication of multiple concomitant infections also emerging periodically. This somehow explained the periods of maintenance of macrolides. In addition, azithromycin was maintained given its anti-inflammatory effects demonstrated in patients with CF and *P. aeruginosa* chronic bronchial infection[14] and the usage of trimethoprim-sulfamethoxazole (without evidence of activity against *M. abscessus*) and levofloxacin, whereas other drugs with proven activity (imipenem, linezolid, and tigecycline) were not tried. Despite levofloxacin is not the optimal therapy, there are some references suggesting its potential use[15] and some *M. abscessus* treatment guidelines in CF include moxifloxacin[16], with levofloxacin being used for *M. abscessus* treatment in CF patients who have poor tolerance to inhaled amikacin.

In the long-term infection in patient 14, we observed the greatest complexity of all the cases in the study, identifying twelve different genotypes, most of them with large pairwise SNV distances between them. These findings could not be due to the usage of an inadequate reference, because for this patient we made the effort to obtain a hybrid assembly from the first genotype identified along her infection, to be used as a reference, to assure the highest precision in calling SNVs. In this case, we were fortunate to have the opportunity to expand our observation window to 16 years, due to the long-term chronic infection of the patient, which could explain the greater complexity found. Another study, taking advantage of a relatively long observation period of 4.5 years, also reported wide diversity among sequential intrapatient isolates, with non-synonymous mutations in 53 different genes[9]. High diversity has also been reported after lung transplantation, even with a narrow observation period, by analyzing isolates not only from sputum specimens, but also from other samples from the respiratory tract[10].

Based on the number of SNVs identified between genotypes, the first interpretation of the findings in our complex case would be a

combination of persistence and sequential reinfections involving several different strains (most of them above the threshold of 25 SNPs generally applied). Using this criterion her disease would be divided into i) a first period of 58 months post-diagnosis, represented by the persistence of one strain with low sequential acquisition of diversity, and ii) a second stage, which started with the identification of a second genotype, intermittently detected, while 10 reinfections with different transient strains followed each other in sequence until the present day. A similar description of a first, clonally homogeneous period, followed by a more complex stage 18 months after diagnosis has also been described elsewhere[9].

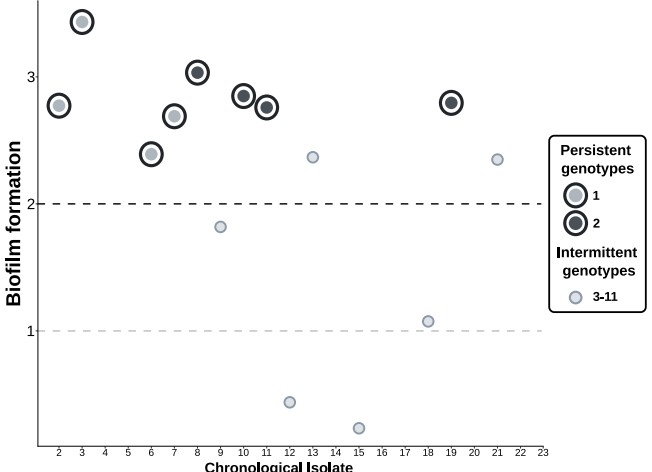

**Fig. 3 | Graphical representation, in chronological order, of the level of biofilm formation for each available isolate.** Values below 1 correspond to non-biofilm formers, values between 1 and 2 are weak biofilm formers, between 2 and 4 moderate biofilm formation.

Our findings were so unexpected that we decided to confirm them by repeating all the sequencing reactions, and even ruling out potential specimen mishandling by demonstrating, with host genetic analysis, that all specimens belonged to the same patient. Nothing in the clinical history of the patient offered any clues as to the reasons for this dramatic shift in the nature of her *M. abscessus* infection. Because her infection was longstanding, she had been seen at two different institutions (her local regional hospital and the national reference center for CF patients); nevertheless, the periods when she interacted with these two different contexts were not associated with the changes we observed over the course of her infection. In contrast, there was one period in January 2018 when her clinical condition worsened and 3 different genotypes were identified within 28 days, which could indicate that there was some kind of relationship between the two observations.

Apart from this interpretation of the genomic data in our patient, we also considered an alternative explanation when we included other relevant findings, and not only the SNP distances between genotypes in relation to reference thresholds. The first was the deep private branch in which all Patient 14 genotypes were located when constructing a phylogenetic tree together with a high number of sequences belonging to the same DCC1. This supported the existence of clear evolutionary relationships between them. Moreover, a Bayesian time tree showed how all Patient 14 genotypes evolved from a common ancestor which corresponded to Genotype 1.

This alternative explanation for patient 14, namely, an infection evolving from a common ancestor that acquired an unexpectedly high diversity, was reinforced by another finding. This was the presence of resistance mutation A2270C in the *rrl* gene in all sequential specimens once it emerged 22 months after the onset of infection. In addition, the same two plasmids were detected in all genotypes, regardless of the genotype involved, except for genotypes 9 and 12 (sharing a terminal branch in the Bayesian tree), which have an incomplete plasmid 1. A series of several reinfections with different resistant strains, all sharing the same mutation and the same plasmids, is highly unlikely.

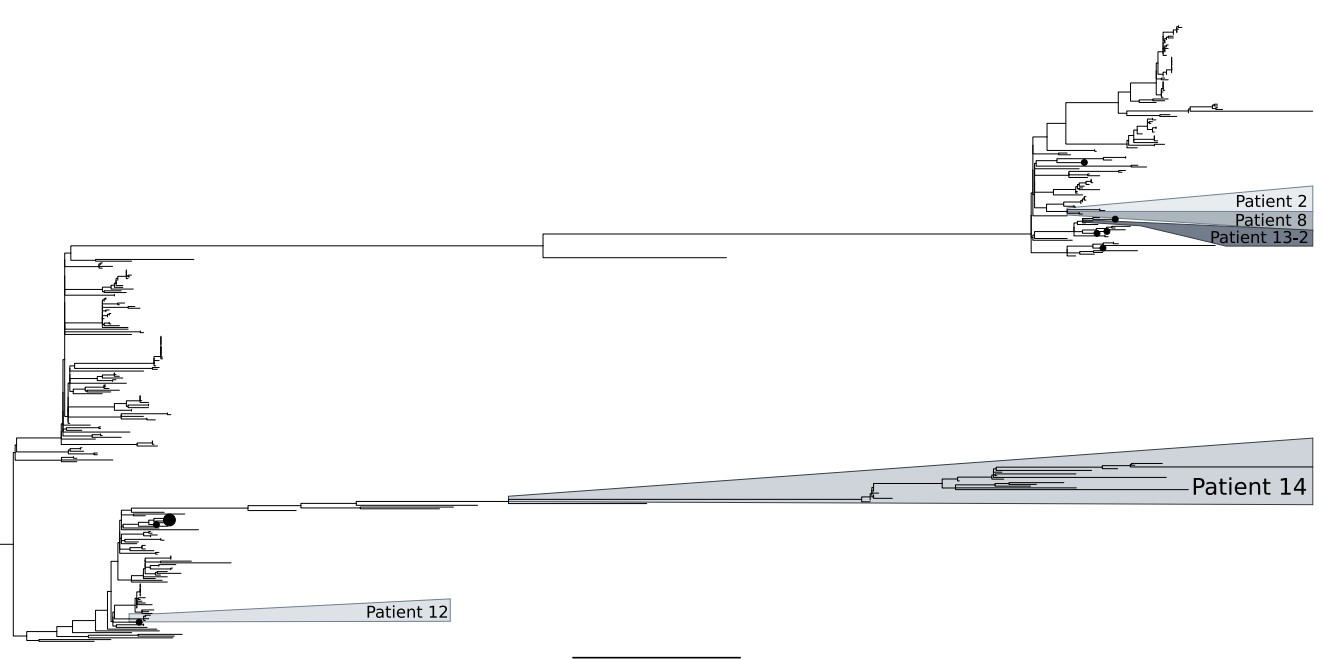

**Fig. 4 | Phylogenetic tree for the *M. abscessus subsp. abscessus* sequences from Patient 14, together with the other sequences from the four patients in our study involving the same DCC1 and another 362 DCC1 representative sequences.** Black dots correspond to the sequences available for other studies in Spain. Scale corresponds to nucleotide divergence (SNPs/site).

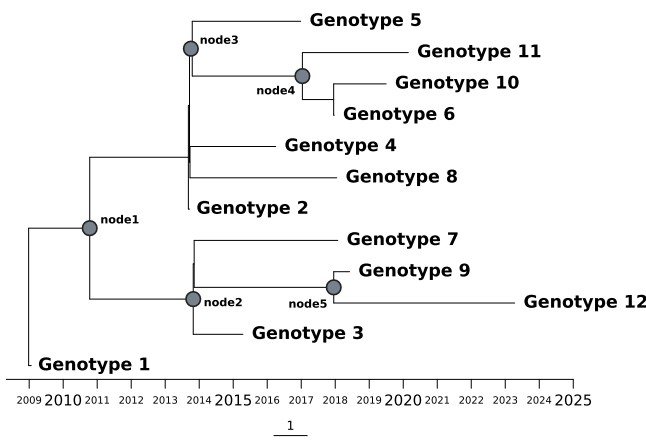

**Fig. 5 | Dated phylogeny of Patient 14 genotypes illustrating the evolutionary relationships and divergence times among the different genotypes from Patient 14.** Branch lengths are proportional to time and nodes represent estimated divergence dates. Genotype 1 corresponds to the third isolate (the first one with the fixed clarithromycin resistance).

Therefore, putting all these observations together, the most likely explanation for this patient was the involvement in this patient's infection of a wide diversity of subclones evolving from a common ancestor that acquired resistance as the result of exposure to antibiotics at the onset of infection. The genetic diversity between genotypes in our patient is above the SNP threshold of 25 (mean pairwise distances 255), but still well below the number of SNVs found in the only clear case of re-infection in our study (15,956 SNVs) and also below the interpatient differences found among strains from the rest of the cases in our study infected with *M. abscessus subsp. abscessus* (an average of 16,357 SNVs). This means that the genomic distances between sequential genotypes in our complex patient are intermediate between the values normally considered for either microevolutions or reinfections. This intermediate diversity could be the result of an unexpectedly high acquisition of diversity in the course of a prolonged chronic infection, rather than reinfections with independent strains. Persistent *M. abscessus* infections have been described as needing to adapt to the changing environment of the lung, and even hypermutator phenotypes have been postulated to meet the adaptive challenge facing this species[6,9]. Genes involved in mutagenesis, endonucleases, peroxidases, and other more specific genes, such as *katG*, *nucS*, *dut*, *ung*, *uvrD* and *dnaE2*, have been studied as possible contributors to these hypermutator phenotypes[6,17].

In patient 14 we detected a pronounced bias in the mutation spectrum, suggesting the involvement of a hypermutator phenotype (Supplementary Data 1); notably, 1147 (98%) of the 1168 substitutions identified in the different genotypes (using as a reference the genotype 1 sequence) corresponded to transitions (enriched in changes TA-CG (71% of the transitions). Similar skewed mutation spectra, towards an enrichment of transitions, have been considered strong evidence for hypermutation in mycobacteria, due to a defective repair system[17]. An indel and non-synonymous change in genes associated to hypermutator phenotypes were identified in all genotypes except in genotype 1 and other additional non-synonymous variants were also identified for additional genotypes. Finally, two genes from the HNH endonuclease family were involved in gene rearrangement events not present in the ATCC reference strain. Therefore, these finding might offer support to explain the burst of diversity and the skewed mutations spectra identified in this patient.

In a long-term infection such as this, a broad set of variants would be more likely than expansion of clonal dominant variants[9]. The detailed analysis of the higher-than-average HZ calls in several of our specimens was consistent with the simultaneous presence of different genotypes identified at other times during infection. Furthermore, a proportion of the HZ calls in these genotypes may correspond to related genotypes that were not independently identified during infection, but shared the same backbone SNVs as some of those that were identified. All the data taken together suggest high complexity in the lung that is not always captured by each sputum sample. Moreover, when we observe the position in the Bayesian time tree of the different genotypes detected along the infection in Patient 14 there is not a correlation between the order in which the genotypes are sampled and their evolutive order. This suggests the existence of a pool of evolved genotypes in lung, likely heterogeneously distributed, which would explain that certain genotypes were randomly sampled at each sampling date (maybe influenced by the clonal composition of the lung site draining to the clinical specimen at each sampling date). Similar complexity in the lung has been found in other studies and different respiratory specimens are required to reveal it[10,18]. Of note, we observed different genotypes in a sputum sample and a BAL specimen taken 9 days apart, indicating compartmentalized infection and highlighting the limitations of using sputum alone to detect the full complexity of the respiratory site.

If this scenario of long-term persistence during which various subclones emerged, we would expect progressive adaptations of the evolved genotypes in the course of the infection. In terms of biofilm formation, we observed no tendency to increased biofilm production over the infection timeline, although the two genotypes that were detected for prolonged periods were significantly higher biofilm producers than those only transiently detected. This could be interpreted as an adaptive evolution involving biofilm development as a mechanism of resistance against antibiotic treatment and immune mechanisms. Since this property is a well-known pathogenic mechanism in this mycobacterium[19], it may explain the persistence of clones as opposed to transient isolates.

Paradoxically, while shorter genomic distances than expected are found between cases from different institutions in the same country, and even different countries[4,20], the intrapatient genomic distances identified in our case even exceeded some of the interpatient distances found in those studies. These findings caution against using strict genomic thresholds to infer interpatient transmission. It has been found that intrapatient microevolution has an impact on the inference of tuberculosis transmission if strict thresholds are applied to determine relatedness[21].

In summary, our study adds new data to the knowledge available on intrapatient diversity in prolonged *M. abscessus* infections. Genomic analysis on sequential isolates indicated that the profile of most of the patients corresponded to persistence with a single strain and moderate acquisition of diversity by microevolution during the infection period, although reinfections with a completely different strain did also occur. One of our cases, a CF patient with chronic infection who was also the longest infected revealed a highly complex pattern of infection, combining the presence of genotypes associated with persistence with the identification of highly diverse sporadic genotypes, some of which coexisted in the same specimen, but maintaining phylogenetic relationships with each other. Our findings challenge the criteria generally used to differentiate between persistence and reinfection, as well as the application of strict diversity thresholds to define relatedness between *M. abscessus* isolates.

## Methods
### Ethical statements
Ethical approval for this study was obtained from the clinical research ethics committee from Hospital 12 de Octubre, in accordance with all relevant ethical regulations. All procedures performed in this study involving human participants were in accordance with the ethical standards of the institutional research committee and with the 1964

**Table 2 | Analysis of the HZs calls identified in Patient 14 genotypes**

| Genotype | 1 | 2 | 3 | 2.1 | 2.2 | 4 | 5 | 2.3 | 6 | 7 | 8 | 9 | 2.4 | 10 | 11 | 2.5 | 12 |
| Number of HZs | 34 | 63 | 55 | 32 | 230 | 10 | 15 | 187 | 104 | 39 | 70 | 15 | 305 | 35 | 63 | 390 | 61 |
|---|---|---|---|---|---|---|---|---|---|---|---|---|---|---|---|---|---|
| Genotype 2 | 0 | – | 0 | – | 33 | – | – | – | – | 0 | – | 0 | 28 | – | – | 20 | 0 |
| Genotype 3 | 0 | 0 | – | 0 | 0 | 0 | 0 | 0 | 0 | 0 | 0 | 0 | 0 | 0 | 0 | 0 | 0 |
| Genotype 4 | 0 | 0 | 0 | 0 | 0 | – | 0 | 0 | 0 | 0 | 0 | 0 | 0 | 0 | 0 | 0 | 0 |
| Genotype 5 | 0 | 0 | 0 | 0 | 0 | 0 | – | 20 | 0 | 0 | 0 | 0 | 0 | 0 | 0 | 0 | 0 |
| Genotype 6 | 0 | 0 | 0 | 0 | 0 | 0 | 0 | 0 | – | 0 | 0 | 0 | 0 | – | 0 | 0 | 0 |
| Genotype 7 | 0 | 0 | 0 | 0 | 0 | 0 | 0 | 0 | 0 | – | 0 | 0 | 0 | 0 | 0 | 0 | 0 |
| Genotype 8 | 0 | 6 | 0 | 17 | 3 | 0 | 0 | 0 | 0 | 0 | – | 0 | 0 | 0 | 0 | 0 | 0 |
| Genotype 9 | 0 | 0 | 0 | 0 | 0 | 0 | 0 | 0 | 0 | 0 | 0 | – | 0 | 0 | 0 | 0 | 1 |
| Genotype 10 | 0 | 0 | 0 | 0 | 0 | 0 | 0 | 0 | 24 | 1 | 0 | 0 | 26 | – | 1 | 55 | 0 |
| Genotype 11 | 0 | 0 | 0 | 0 | 0 | 0 | 0 | 0 | 0 | 0 | 0 | 0 | 0 | 0 | – | 0 | 0 |
| Genotype 12 | 0 | 0 | 0 | 0 | 0 | 0 | 0 | 0 | 0 | 0 | 0 | 0 | 0 | 0 | 0 | 0 | – |
| Node 1 | 5 | – | – | – | – | – | – | – | – | – | – | – | – | – | – | – | – |
| Node 2 | 0 | 5 | – | 0 | 35 | 0 | 0 | 0 | 0 | – | 0 | – | 27 | 0 | 0 | 28 | – |
| Node 3 | 0 | 0 | 0 | 0 | 0 | 0 | – | 0 | – | 0 | 0 | 0 | 6 | – | – | 5 | 0 |
| Node 4 | 5 | 1 | 5 | 0 | 0 | 0 | 0 | 5 | – | 6 | 5 | 0 | 117 | – | – | 78 | 0 |
| Node 5 | 3 | 5 | 6 | 4 | 6 | 3 | 5 | 5 | 5 | 6 | 5 | – | 20 | 5 | 5 | 75 | – |
| Matched alleles | 13 | 17 | 11 | 21 | 77 | 3 | 5 | 30 | 29 | 13 | 10 | 0 | 224 | 5 | 6 | 261 | 1 |
| % of matched alleles | 38.24 | 26.98 | 20.00 | **65.63** | 33.48 | 30.00 | 33.33 | 16.04 | 27.88 | 33.33 | 14.29 | 0.00 | **73.44** | 14.29 | 9.52 | **66.92** | 1.64 |

Number and % of the alleles involved in the HZs calls for each genotype that corresponds to the SNPs identified in the other genotypes or nodes (according to the tree in Fig. 5). In bold are highlighted the three highest percentages of alleles from HZ calls which matched with SNPs from other genotypes.

Helsinki declaration and its later amendments. Informed consent was obtained to use sociodemographic data from the patients.

## Short-read sequencing

Whole genome sequencing was performed on genomic DNA purified from culture isolates. The isolates were inactivated and purified following the manufacturer's instructions (Qiagen kit; QIAamp mini-DNA kit; QIAGEN, Courtaboeuf, France). Libraries were prepared using the Nextera XT kit (Illumina) following the manufacturer's instructions, and were run on a MiSeq device (2 x 151bp), which resulted in an average per-base coverage of 85.72X, with 94.14% of the genome at >20X coverage.

Sequence analysis was done using an in-house pipeline deposited in Git-Hub: https://github.com/MG-IiSGM/autosnippy[22]. Briefly, the pipeline performed the following steps: i) species identification, with Kraken2 v2.1.2 and Mash v2.3; ii) mapping and variant calling used snippy v4.6.0; Burrows-Wheeler Aligner (BWA-MEM v0.7.17) was used for mapping, and Freebayes v1.3.2 for variant calling, using *Mycobacterium abscessus* ATCC 19977 (CU458896.1) or *Mycobacterium abscessus subsp. massiliense* str. GO 06 (NC_018150.2) as references, for variant calling in Patient 14 genotypes, an assembly of Genotype 1 was obtained to be used as the reference (see the section Short and long-read hybrid assembly for details); iii) variant annotation used the SnpEff v5.1 tool; and iv) recalibration of occasional low-coverage positions using joint variant calling. Highly polymorphic and repetitive regions, phages, and PE/PPE regions were removed from the final SNV distance calculation and annotation. SNVs located in areas with a higher-than-expected number of calls (≥3 SNVs within 10 bp of each other) were also excluded.

Alignments and SNV variants were visualized and checked with the IGV (Integrative Genomics Viewer) program. Different genotypes were considered when the number of SNPs between the sequences exceeded the genomic threshold (25-30 SNPs) applied in the literature to consider two *M. abscessus* isolates as different strains[3,4,10,11].

## Short-read assembly

Fastq files were pre-processed with fastp v0.23.2. This involved quality filtering by applying a Phred score >Q30, a minimum length of 35 base pairs, and an additional parameter for adapter detection in paired-end reads. Subsequently, short-read assemblies were performed by using Unicycler v0.5.0 with default parameters.

**Long-read sequencing.** Libraries were prepared from purified DNA, using the Ligation sequencing gDNA - native barcoding protocol (SQK-LSK109, ONT, Oxford, UK), following the manufacturer's instructions. Barcoding and adapter ligation were performed with the Native Barcoding Expansion kit (EXP-NBD114) and a total of about 6 ng/µL of each of the final libraries was loaded into a MinION flow cell (R.9.4.1 FLO-MIN106).

An in-house pipeline, https://github.com/MG-IiSGM/prokaION[23], was applied to analyze the sequencing reads. Briefly, the pipeline went through following steps: i) preprocessing fast5 files into fastq format, including basecalling (the dna_r9.4.1_450bps_hac configuration was followed, using the GPU device); ii) barcode demultiplexing, identifying barcodes, adapters and potential primers to remove them, using Guppy v.6.4.6; iii) quality assessment (mean Phred score >Q10) and trimming of 20 base pairs from both ends of the sequence with Chopper v0.7.0; and iv) taxonomic identification with Kraken2 v2.1.2 and Mash v2.3.

**Short and long-read hybrid assembly.** For genotype 1 from Patient 14, once the fastq files with quality reads were trimmed, filtered according to base quality and fragment length, hybrid assembly of both long and short reads was performed. Firstly, a hybrid assembly was performed by using Unicycler v0.5.0, with short and long-read

fastq files and then removing <1000pb contigs. A second assembly was performed by using Flye v2.9, from only high-quality filtered long-read fastq file. Subsequently, assembly polishing was performed with Medaka v1.11.2 for long reads and Polypolish v0.5.0 for short reads. A consensus between the two assemblies was generated with Trycycler v0.5.4. Resulting in a 1-contig assembly for the chromosome of 5,521,121.

## Plasmid analysis

From the fasta file obtained from the different assemblies (either for short-read or hybrid assemblies), plasmid detection was performed using two approaches: i) First, PlasmidID v1.6.4 was initially executed with its default database including all curated plasmids from the RefSeq collection at NCBI; ii) subsequently, a blastn v2.14.1+ search was conducted against the full nucleotide (nt) database to assign contigs corresponding to plasmids.

## Phylogenetic analysis

Additional sequences were extracted from Ruis et al.[12] (Source Data, Phylogeny_SRA.xlsx) which encompassed 1335 and 710 sequences from *subsp. abscessus* and *subsp. massiliense*, respectively, categorized in DCCs and FastBAPS clusters. To ensure representativeness, 15 sequences for each DCC were randomly selected, with the inclusion of all sequences from Spain as a prerequisite. In addition, another 15 sequences from each FastBAPS cluster were included (when represented by fewer than 10 sequences all were included). Therefore, for the phylogenies, 252 *subsp. abscessus* sequences and 150 from *subsp massiliense*, were included, together with the sequences from our study.

**Subspecies phylogeny.** For each of the two subspecies involved in our study, *abscessus* and *massiliense*, a core SNP analysis using snippy-core (included in autosnippy) was performed. Then, a core SNP alignment was generated, to construct a high-resolution phylogeny, ruling out potential recombination events.

The phylogenetic trees were constructed using the RAxML v8.2.12 (Randomized Axelerated Maximum Likelihood) software with the General Time Reversible (GTR) nucleotide substitution model and GAMMA rate heterogeneity. Additionally, a total of 1000 starting trees were integrated, applying a seed '12345' to ensure result reproducibility. Phylogenetic trees were visualized using Figtree v1.4.4, with a root tree positioned at the midpoint.

**Dominant circulating clone phylogeny.** A specific phylogenetic analysis was done for the Patient 14 genotypes, including additional representatives of the same Dominant Circulating Clone (DCC1). All 362 FastQ *RR* DCC1 sequences from the SRA database (Ruis et al.[12]) were included in this analysis. The procedures followed in the precedent section were also followed here, but now using a whole-genome SNP alignment, to compensate the lower diversity expected due to restricting the analysis to DCC1. Phylogeny was visualized with Figtree v1.4.4 with a root tree at midpoint, arranged in ascending node order.

**Temporal phylogeny of Patient 14.** To assess the temporal evolutionary phylogeny of Patient 14 genotypes, once obtained the high-quality whole-genome SNVs alignment with respect to Genotype 1 assembly, we reconstructed the temporal evolution by using BEAST v1.10[24]. The substitution model employed was GTR, with an empirical frequency base and a GAMMA heterogeneity model. A relaxed clock model was employed, under a coalescent tree prior, Bayesian skyline, using 200 million MCMC (Markov Chain Monte Carlo) stages.

To ensure reproducibility, the model was replicated three additional times, verifying parameter consistency, and a consensus tree for the entire dataset was obtained.

## Host genetic analysis

Short tandem repeat STR-PCR (Mentype® Chimera® Biotype, Germany) was applied for human identity testing and analysis, using the same specimens that had been used for genome sequencing. Twelve non-coding STR loci and the gender-specific amelogenin locus, labeled with three different dyes (6-FAM™, BTG, or BTY) were examined. PCR was performed with 0.2–1 ng of genomic DNA, using the Mentype® Chimera® PCR amplification kit (Biotype, Germany), the GeneAmp® PCR System 9700 Thermal Cycler (Applied Biosystems), followed by capillary electrophoresis in the 3130*xl* Genetic Analyzer (Applied Biosystems), as recommended by the manufacturer.

## Biofilm formation

Biofilm was quantified following a modified version of the method described by Stepanovic et al.[25]. The amount of mycobacterial culture was calculated to prepare a 10 mL PBS suspension with an optical density of 0.1 at 600 nm, and 100 μL was deposited in the wells of a non-treated P96 plate, left to incubate for 90 min at 37 °C, followed by two washes in 100 μL PBS solution. 100 μL of Middlebrook 7H9 medium was added to the wells and the plates were incubated for 4 days at 30 °C. The supernatant was removed, 50 μL of 2% crystal violet was added and incubated for 20 min. Excess crystal violet was removed, and the dye was solubilized with 100 μL of absolute ethanol. Finally, absorbance was measured at 570 nm.

## Statistical analysis

Pearson's correlation coefficient was used to assess linear associations between different isolates. The Tukey test enabled multiple comparisons of means between groups, while the Welch t-test compared means between groups with unequal variances. Statistical significance was set at $p < 0.05$ for all analyses.

## Reporting summary

Further information on research design is available in the Nature Portfolio Reporting Summary linked to this article.

## Data availability

The data supporting the results of this study (FastQ files and assembly) were deposited in the ENA (https://www.ebi.ac.uk) under project accession number PRJEB61124. Source data are provided with this paper.

## Code availability

The code used is accessible through the research group's public repository on GitHub: Microbial Genomics – Gregorio Marañón Health Research Institute, or via the link or DOI provided in the corresponding Methods section.

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

## Acknowledgements

This work was supported by the *Instituto de Salud Carlos III* [AC16/00057, FIS15/01554, PI21/01823, PI19/00331, FI20/00129, PI21/01738],

co-financed by European Regional Development Funds of the European Commission: "A way of making Europe"; a Miguel Servet Contract (ISCIII) CPII20/00001 to LPL. FI22/00145 contract from a PFIS (ISCIII) to SBS and Ministerio de Ciencia (MCIN/AEI/10.13039/501100011033, grant PID2020-112865RB-I00). The authors are grateful to Adrian M. Zelazny from the NIH Clinical Center America's Research Hospital for discussions around hypermutators and for providing us with a list of genes potentially involved in this phenotype, to Iñaki Comas and Ana María García-Marín from the Instituto de Biomedicina de Valencia (IBV-CSIC) for discussions about the hybrid assembly procedures and to Janet Dawson for editing and proofreading assistance. The authors would like to express their sincere gratitude to EGF for the participation, collaboration and willingness to engage in interviews, which greatly contributed to a better understanding of their specific case.

## Author contributions

The conception and design of the study were carried out by L.P.L. and D.G.V. M.M.L., J.E., C.T., P.L.R., D.D., A.R.L., M.J.R.S. and E.Z.d.L. were responsible for the collection of clinical specimens and/or patient data. M.H.M., A.B.L., A.M.S. and A.S.P. performed experiments. Data acquisition and bioinformatics analysis was performed by S.B.S. The interpretation of the data was carried out by S.B.S., L.P.L., J.B. and D.G.V. Visualization of the results was done by S.B.S. The manuscript written was done by S.B.S., P.M., L.P.L. and D.G.V. Finally, all authors critically reviewed and provided their final approval for the manuscript's submission.

## Competing interests

The authors declare no competing interests.

## Additional information

[1]Servicio de Microbiología Clínica y Enfermedades Infecciosas, Hospital General Universitario Gregorio Marañón, C/Doctor Esquerdo, 46, 28007 Madrid, Spain. [2]Instituto de Investigación Sanitaria Gregorio Marañón (IiSGM), C/Doctor Esquerdo, 46, 28007 Madrid, Spain. [3]Escuela de Doctorado, Universidad de Alcalá, Plaza de San Diego, s/n, 28801 Alcalá de Henares, Madrid, Spain. [4]Unidad de Gestión de Laboratorio, Hospital Universitario Torrecárdenas, Almería, Spain. [5]Servicio de Microbiología, Instituto de Investigación Sanitaria Fundación Jiménez Díaz-UAM, Hospital Universitario La Fundación Jiménez Díaz, Av. de los Reyes Católicos, 28040 Madrid, Spain. [6]Centro de Investigación Biomédica en Red (CIBER) de Enfermedades Infecciosas - CIBERINFEC, Instituto de Salud Carlos III, Madrid, Spain. [7]Department of Microbial Biotechnology, National Center for Biotechnology, Consejo Superior de Investigaciones Científicas (CSIC), C/ Darwin, 3, Campus de la Universidad Autónoma-Cantoblanco, 28049 Madrid, Spain. [8]Microbiología y Enfermedades Infecciosas, Hospital Universitario 12 de Octubre, Av. de Córdoba, s/n, 28041 Madrid, Spain. [9]Servicio de Microbiología y Parasitología, Hospital Universitario La Paz - IdiPAZ, Madrid, Spain. [10]Servicio de Microbiología, Instituto de Investigación Sanitaria, Hospital Universitario La Princesa, Calle de Diego de León, 62, 28006 Madrid, Spain. [11]Servicio de Neumología, Hospital Universitario La Paz -IdiPAZ, Madrid, Spain. [12]Centro de Investigación Biomédica en Red (CIBER) de Enfermedades Respiratorias - CIBERES, Instituto de Salud Carlos III, Madrid, Spain. [13]Departamento de Medicina, Universidad Complutense, Av. Séneca, 2, 28040 Madrid, Spain. [14]These authors contributed equally: Laura Pérez-Lago, Darío García de Viedma. ✉e-mail: lperezg00@gmail.com; dgviedma2@gmail.com

