## [Peer Review File · Nature Communications]

REVIEWER COMMENTS

Reviewer #1 (Remarks to the Author):

The manuscript by Buenestado-Serrano et al describes an in depth analysis of M abscessus infections in a population, with particular investigation of a specific chronic infection. Overall the paper is clear and concise and the strong discussion section wraps it up nicely. I only have some minor points to address:

- I think the discussion states well the likely scenario of rapid evolution of a single strain in patient 14 but this is not as clear in the results. It would be good if perhaps a SNP distance matrix or some illustration of share distances (perhaps on figure 4) was added to see how related all the strains are to each other. Are there many shared SNPs that would be of note, such as drug resistance mutations (beyond the one they all have)? How do you explain early genotypes being more related to later ones than sequential ones (e.g. 3 and 7 seem more related).
 - a. As a note here, a Bayesian time tree may serve better than the network, given the distances and would make it easier to see the sequential evolution, if that is the case.
- The introduction could use a little more background on why we need to know the inpatient diversity. What clinical benefit does it bring? The knowledge gap is not very evident beyond 'we want to know more' (it is covered in the discussion but good if the reader understands better at the outset).
- Line 210: I think you mean patient 14, not 4?
- Table 1: Could accession numbers for the isolates be added to this table? It may make it too crowded but just a suggestion for ease of access to data.
- Species spelled incorrectly in Table 1

Reviewer #2 (Remarks to the Author):

This manuscript describes an analysis of intra-patient variation in chronic Mycobacterium abscessus infections. Most of the work focuses on patient 14, for whom multiple related genotypes were identified.

Comments:

- 1) The authors emphasise in several places that analyses on the intra-patient diversity of M. abscessus is scarce. I disagree - there are multiple works looking at this in quite a lot of detail, and much is known about the rate and types of mutations that occur during chronic infections within this species (primarily ref 14). In this study the sample numbers are small and the analysis limited. Therefore I don't think this work is a significant advance on our knowledge of M. abscessus intra-patient diversity.
- 2) The phylogenetic trees do not have any scale bars. And were constructed using a very simplistic method. Maximum likelihood methods are usually recommended when constructing bacterial WGS trees.
- 3) Most of the paper focuses on the high SNP distances found in patient 14. The distances and phylogenetic relatedness suggest that these are related genotypes. In order to test this is not the result

of reinfection the authors need to combine their data with publicly available *M. abscessus* sequences

4) The authors suggest that one possible cause for the high number of related genotypes in patient 14 is hypermutation. This possibility doesn't appear to have been investigated in detail despite the fact this has been documented previously (refs 1 and 14). They mentioned that mutations were not found in repair genes - but the authors need to confirm if they have looked for indels or larger rearrangements. In addition I recommend that the authors investigate the mutation spectra in this patient - as a skewed mutation spectra is strong evidence for hypermutation due to a defective repair system.

Reviewer #3 (Remarks to the Author):

Comments for the Authors

In the submitted manuscript, the authors use a comparative genomics approach to analyze sequential isolates of *Mycobacterium abscessus* (Mab) associated with 14 Patients (4 with Cystic Fibrosis). Although chronic infections (and sometimes re-infections) are common in this patient population, only a few studies have examined intra-patient diversity and microevolution of Mab. In the current study, the comparative genomics approach definitively identified at least one case of re-infection (Patient 13). However, the results suggest that, for the majority of infections, there were few changes between paired isolates, even after 2 months to several years of persistent infection. The major focus was on the case of Patient 14 who had 23 follow up isolates of *M. abscessus* subsp. *abscessus* over a 16 year period. The detailed analysis of those isolates highlights the complexity of long term Mab infections.

Major concerns include

1. Limitations of the bioinformatics methods
2. Limitations of the SNP analysis
3. Limitations of the plasmid analysis
4. Limitations in the presentation of the phylogenetic data

Bioinformatics Methods and SNP Analyses;

The Methods should include additional details, such as the specific settings used with the various sequencing analysis tools.

According to lines 358-369, a reference mapping approach was used. The Methods state that "Highly polymorphic and repetitive regions, phages, and PE/PPE regions were removed from the final SNP distance calculation and annotation" (Lines 336-337), but the Results don't indicate what proportion of the reference genome was actually used or what proportion of the reads from each Patient isolate were discarded or leftover. The removal of those sequences, as well as the exclusion of "SNVs located in areas with a higher-than-expected number of calls", suggests that the reported SNP distances are likely an underestimate of the absolute number of differences.

The Methods indicate that SNP distances were calculated in comparison to the genome of the Mab Type strain ATCC 19977. However, mapping to a project-specific reference genome should provide a more accurate measure of SNP differences. Lines 389-391 indicate that hybrid assemblies were constructed for some isolates of Patient 14. Especially for the case of Patient 14, reference mapping to the hybrid assemblies derived from Genotype 1 and/or Genotype 2 datasets should allow more robust mapping of

SNVs, in/dels and genomic rearrangement events, and improve characterization of the relationship between the numerous isolates and genotypes.

Much of the Discussion is speculation about the relationships between the various Patient 14 genotypes. The presence of the 'fixed' A2270C mutation, the presence of two identical plasmids and the rough colony type associated with the 20 bp mps2 insertion across multiple genotypes is consistent with an infection evolving from a common ancestor since these findings (except for the plasmids) are rarely seen in first time unrelated isolates. The authors should determine if an erm (41) gene with the T28C substitution is present in the first, macrolide susceptible Genotype 1 strain as well as all the subsequent Patient 14 isolates of genotypes 1-12. In Bronson et al (ref 2), the Mab Type strain ATCC 19977 is assigned to Clade A1. In contrast, the T28C mutation is associated with a distinct set of circulating strains, Clade A3. The A2270C mutation accounts for 'reversion' to a macrolide resistant phenotype. However, the presence of T28C may provide clade information that determines the need to repeat SNV analyses with a more closely related (e.g., Clade A3) reference genome instead of ATCC 19977 (Clade A1). In general, the manuscript would benefit from additional details about the SNPs – including, but not limited to, the numbers of non-synonymous mutations, genes impacted by the nonsynonymous SNPs, and any association of the SNPs with specific genomic changes (in/dels, transposons, etc). Are the SNPs in 'core genes' or the accessory genome?

Plasmid Analysis

The Abstract states that part of the purpose of the study was to characterize plasmids. It appears that plasmid analysis was only performed for isolates from Patient 14. If plasmid analysis wasn't performed for at least one isolate from every Patient, the authors should consider screening those datasets for sequences matching plasmid 1 (pJCM30620_1) and/or plasmid 2 (pS2c). Plasmids have only rarely been utilized as part of the genomic comparisons in mycobacteria and the additional plasmid analysis – as well as more discussion about these findings - would provide valuable information about the distribution of plasmids across Mab strains.

In line 210, the authors refer to patient 4. This is likely a typo and should be 14.

Phylogenetic Analysis

Figure 1 is a good depiction of the phylogeny of the sequenced isolates. However, it would be helpful to include lengths (# SNVs) for each branch as well as labels for each Patient 14 isolate (e.g., 14-1, 14-3).

Figure 4 should also include branch lengths. Consider combining the data from Tables 2 and 3 into a single table, or displaying that SNP and HZ data in Figure 4.

Do the various genotypes and phylogenetic clusters identified in the current study correlate with any of the clades described by Bronson et al (ref 2) Or sequence types from the Mab MLST scheme? (e.g., <https://pubmlst.org/organisms/mycobacteroides-abscessus-complex/>)

Additional comments

Table 1: Remove the comment from Patient 14 that findings reflect "Mixed events (Persistence and Reinfection)" and change to "Persistence and Microevolution" since that is your conclusion presented as an "alternate conclusion" on page 13 of the Discussion.

Do the hybrid assemblies include additional chromosomal sequences that didn't map to the ATCC 19977 reference genome? Although repetitive sequences and PE/PPE genes can compromise reference

mapping, the presence/absence of specific prophage sequences could be phylogenetically informative.

Clinical Summary

This paper is not about drug therapy of Mab and initial therapy was more than 15 years ago. However, since others will read about this therapy, some comments about its inadequacies need to be pointed out. Not to be overly critical, but only to be sure that a constructive and correct message is provided. Two of the drugs (TMP/SMX and fluoroquinolones) have no proven activity against Mab, whereas other drugs with proven activity (imipenem, linezolid, and tigecycline and now other better tolerated drugs omadacycline and eravacycline though the latter only recently available) were not reported as having been tried. In a species like Mab, which has only a single copy of the ribosomal rRNA operon, it's not unexpected that prolonged macrolide monotherapy resulted in the development of mutational resistance. With macrolide susceptible isolates, we now know that appropriate treatment involves a macrolide plus a proven (IV) companion drugs and that prolonged combination therapy comes with a high expectation of cure. The patient did receive cefoxitin and amikacin, but only a month at a time, which is clearly an inadequate duration. The Discussion should include additional comments about therapy, especially about avoidance of macrolide monotherapy and (for infections with macrolide susceptible Mab) promotion of combination therapy with macrolides plus proven companion (IV) meds.

Reviewers Conclusions:

What are the noteworthy results:

- (1) Microevolutionary comparison of isolates from persistent infections, especially the very long term follow up (16 years) of isolates from Patient 14.
- (2) Demonstration of the utility of plasmid analysis as part of genomic comparisons.
- (3) Will the work be of significance to the field and related fields ? Yes. This study adds to our knowledge of genome evolution during Mab infection and highlights the need to analyze multiple isolates per infection.
- (4) How does it compare to the established literature? There is minimal data about genomic analysis with long term follow up of Mab infected patients.
- (5) Does the work support the conclusions and claims, or is additional evidence needed ? In general, the evidence supports the claims, but additional evidence would lead to more definitive and robust conclusions. For example, the use of a project-specific reference genome could better define the relationships among the Patient 14 isolates and genotypes. Characterization of non-synonymous SNPs might reveal evolutionary trends, provide insight into the emergence of antibiotic resistance (e.g., the role of erm(41) C28T), and enhance the phylogenetic analysis.
- (6) Are there any flaws in the data analysis, interpretation, and conclusions ? As described above, there are some flaws in the data analysis and shortcomings to the interpretation and conclusions.
- (7) Is the methodology sound ? Does the work meet the expected standards in the field ? As described above, some improvements in methodology are required.
- (8) Is there enough detail provided in the methods for the work to be reproduced ? As described above, additional details should be provided.

Reviewer #4 (Remarks to the Author):

Answers to Reviewers' comments

Reviewer #1 (Remarks to the Author):

The manuscript by Buenestado-Serrano et al describes an in depth analysis of *M. abscessus* infections in a population, with particular investigation of a specific chronic infection. Overall the paper is clear and concise and the strong discussion section wraps it up nicely. I only have some **minor points** to address:

- I think the discussion states well the likely scenario of rapid evolution of a single strain in patient 14 but this is not as clear in the results. It would be good if perhaps a SNP distance matrix or some illustration of share distances (perhaps on figure 4) was added to see how related all the strains are to each other. Are there many shared SNPs that would be of note, such as drug resistance mutations (beyond the one they all have)?

We would like to remind that a matrix of SNPs distances for the Patient 14 genotypes had already been included in our MS (Supplementary material in the previous version, now renumbered as Supplementary Table 3 in this new version).

In addition, regarding the reviewer's alternative suggestion of including some illustration of distances on Figure 4, we need to clarify that the network included in Figure 4 in our previous version has been now substituted by a Bayesian tree (as requested by the reviewers, new Figure 5). It means that SNP distances can't be added to the branches as distances in this new tree does not correspond to SNP-based distances.

Nevertheless, another new analysis (a DCC1 phylogeny, new Figure 4) included in this new version as per reviewer's requests, can also be helpful to address this reviewer's point and also to answer to his/her interest "to see how related all the strains are to each other" and, finally, to reinforce our interpretation of a single strain evolving rapidly within patient into different genotypes. In our opinion, this approach is more suitable to address all these points, e than looking for additional relevant SNPs shared by all the evolved genotypes in Patient 14.

*Therefore, we have now integrated in a phylogeny the sequences of all the genotypes from Patient 14 with 362 *M. abscessus* subsp. *abscessus* sequences sharing the DCC (DCC1) (taken from Ruis et al. (2021); this compilation also includes all the 13 available FastQ files from Spain in SRA). From this new analysis it can be observed (Figure 4) how closely related are Patient 14 genotypes between them and how far are from the other *M. abscessus* representatives belonging to the same DCC1. All genotypes from patient 14 share a single, deep, branch in the tree, sharing a common ancestor, and are distant from the other 362 sequences. 102 private SNPs are responsible for this deep branch for Patient 14 genotypes.*

We hope that this analysis is robust enough to address these reviewer's concerns. We have also added new comments in the text regarding this issue (lines 171-176 at the cleaned version or 176-191 at the changes control version).

How do you explain early genotypes being more related to later ones than sequential ones (e.g. 3 and 7 seem more related).

As detailed in Results (in the section “Analysis of coexistence of genotypes”) and in Discussion, we identified a high number of HZs calls in certain genotypes and we found that the alleles involved in the HZ calls correspond to SNPs called in other genotypes. These observations led us to interpret that a pool of different evolved genotypes was coexisting in the patient. Probably, due to the likely different clonal composition in the lung sites draining into the sputum at each sampling date, we detected (somehow randomly) one or another of the coexisting genotypes. This means that the order in which the genotypes are detected in the clinical specimens (early or late sampled genotypes) are likely only a snapshot of the total clonal complexity existing in the lung. These sequential snapshots don’t necessarily correspond to the evolutionary (phylogenetic) order followed by the genotypes to emerge (early or late evolved genotypes). We labelled the genotypes from 1 to 12, just following the chronology in which they were identified from the sequential specimens, but this order does not necessarily reflect the order in which they emerged/evolved. This explains the apparent discrepancies raised by the reviewer. To clarify these aspects, we have now included some of these ideas in the Discussion (lines 376-381 at the cleaned version or 417-422 at the changes control version):

“Moreover, when we observe the position in the Bayesian time tree of the different genotypes detected along the infection in Patient 14 there is not a correlation between the order in which the genotypes are sampled and their evolutive order. This suggests the existence of a pool of evolved genotypes in lung, likely heterogeneously distributed, which would explain that certain genotypes were randomly sampled at each sampling date (maybe influenced by the clonal composition of the lung site draining to the clinical specimen at each sampling date)”.

- a. As a note here, a Bayesian time tree may serve better than the network, given the distances and would make it easier to see the sequential evolution, if that is the case.

Following the reviewer suggestion, we have now substituted our previous network (included in Figure 4) by a Bayesian time tree (new Figure 5). Now branch lengths are measured not in genomic SNP-distances, but in temporal years. This allows us to emphasize the temporal aspects of the evolutionary divergence, which provides a more nuanced perspective on the evolutionary timeline. We fully agree with the reviewer that, now, it is much easier to see the sequential evolution.

We have now specified in Methods the procedures to perform this new analysis (lines 503-511 at the cleaned version or 575-583 at the changes control version) and rephrased the content in Results according to this new analysis (lines 324-326 at the cleaned version or 358-360 at the changes control version).

- The introduction could use a little more background on why we need to know the inpatient diversity. What clinical benefit does it bring? The knowledge gap is not very evident beyond 'we want to know more' (it is covered in the discussion but good if the reader understands better at the outset).

We have now added two new paragraphs in Introduction aiming to complete the need to explore more in detail inpatient diversity (lines 73-78 at the cleaned version or 77-82 at the changes control version).

- Line 210: I think you mean patient 14, not 4?

Regarding the comment on line 210, indeed, it was a typographical error, and we intended to refer to patient 14, not patient 4. The error has been corrected in the manuscript.

- Table 1: Could accession numbers for the isolates be added to this table? It may make it too crowded but just a suggestion for ease of access to data.

We have now added accession numbers to Table 1.

- Species spelled incorrectly in Table 1

The reviewer is correct about the typographical error in the header of column 4 in Table 1. The error has been rectified in the revised Table 1.

Reviewer #2 (Remarks to the Author):

This manuscript describes an analysis of intra-patient variation in chronic *Mycobacterium abscessus* infections. Most of the work focuses on patient 14, for whom multiple related genotypes were identified.

Comments:

1) The authors emphasise in several places that analyses on the intra-patient diversity of *M. abscessus* is scarce. I disagree - there are multiple works looking at this in quite a lot of detail, and much is known about the rate and types of mutations that occur during chronic infections within this species (primarily ref 14). In this study the sample numbers are small and the analysis limited. Therefore I don't think this work is a significant advance on our knowledge of *M. abscessus* intra-patient diversity.

We acknowledge that there are several robust analyses according to this issue and therefore we have modified our statements in the text to eliminate our consideration that these analyses are scarce.

2) The phylogenetic trees do not have any scale bars. And were constructed using a very simplistic method. Maximum likelihood methods are usually recommended when constructing bacterial WGS trees.

We appreciate the reviewer's insightful observation regarding the simplicity of our initial phylogenetic approach, which relied on a genomic distance matrix. We totally agree that an approach based on maximum likelihood methods must be applied to our data.

*Now, two independent Maximum Likelihood trees based on core SNP analysis (Figure 1a and b) have been obtained for the sequences in our study, which corresponded to *Mycobacterium abscessus* subsp. *massiliense* and *Mycobacterium abscessus* subsp. *abscessus*, respectively. To enhance the quality of our subspecies phylogeny, in addition to our sequences we have fed the trees with additional sequences for these subspecies extracted from an exhaustive dataset (Ruis, Christopher, et al. "Dissemination of *Mycobacterium abscessus* via global transmission networks." *Nature Microbiology* 6.10 (2021): 1279-1288), which encompassed 1335 and 710 sequences from the subsp. *abscessus* and *massiliense*, respectively, categorized in DCCs and FastBAPS clusters. To ensure representativeness, we executed a random selection process for each subspecies, with the inclusion of all the sequences from Spain as a prerequisite. We selected 15 sequences for each DCC. In addition, we selected 15 sequences for each FastBAPS cluster (when represented by fewer than 10 sequences all were included).*

*Finally for our new phylogeny, we included 252 sequences for subsp. *abscessus* and 150 from subsp. *massiliense*. The new methodology used for the ML phylogeny is now stated in Methods ('Phylogenetic Analysis', lines 475-511 at the cleaned version or 548-583 at the changes control version).*

3) Most of the paper focuses on the high SNP distances found in patient 14. The distances and phylogenetic relatedness suggest that these are related genotypes. In order to test this is

not the result of reinfection the authors need to combine their data with publicly available *M. abscessus* sequences

Following the reviewer's request, we have performed a phylogenetic analysis (based on whole genome SNP alignment) integrating the genotypes from Patient 14 with another 362 Mycobacterium abscessus subsp. abscessus sequences (taken from Ruis et al. (2021)) belonging to the same DCC than the one involved in the infection of Patient 14 (DCC1). We have also included all the FastQ files (13 sequences) available for Spain for this DCC in SRA. This new tree is included in Figure 4.

From this new analysis, it can be observed that all genotypes from patient 14 share a single, deep, branch in the tree, sharing a common ancestor. 102 private SNPs are responsible for this private deep branch for Patient 14 genotypes. Due to the depth of the Patient 14 branch, all the 362 sequences, including those from Spain, are located too far to consider that reinfection may have a role in Patient 14 findings. We really appreciate the suggestion from the reviewer that offered as the opportunity to reinforce by this analysis the relatedness between Patient 14 genotypes, and to fully rule out the intervention of reinfection in the diversity observed along her infection. We have rewritten accordingly all the "Phylogenetic analysis" sections and the corresponding methods.

4) The authors suggest that one possible cause for the high number of related genotypes in patient 14 is hypermutation. This possibility doesn't appear to have been investigated in detail despite the fact this has been documented previously (refs 1 and 14). They mentioned that mutations were not found in repair genes - but the authors need to confirm if they have looked for indels or larger rearrangements.

As the reviewer 2 properly points out, we hypothesized that the burst of diversity found in patient 14 may be result of some genetic change along the infection, leading to the occurrence of a hypermutator phenotype. However, we must acknowledge, as raised by the reviewer, that an analysis on this matter was not properly performed.

As a result of the reviewer's comment, we have now checked all patient 14 sequential genotypes, looking for mutations and also for indels and rearrangements in 65 genes potentially associated with hypermutation (coding for peroxidases, catalases, genes involved in DNA repair, etc).

A non-synonymous change (Val617Met in MAB_3516c, encoding the uvrD-like helicase protein) was identified in all genotypes except in genotype 1 and an indel (Thr143fs in MAB_3543c, encoding the RNA polymerase sigma-E factor) was also identified in all genotypes and in two representatives of Genotype 1 (isolates 2 and 7).

*In addition, different non-synonymous variants (7) were identified; three in genotype 3 (Pro67Ser in MAB_1160, Asn84Ser in MAB_2712c and Glu145Gly in MAB_3909); one in genotypes 6 and 10 (Arg66His in the MAB_1349 gene) three in genotypes 5, 8 and 11 each (Val217Ala in MAB_3480, Thr109Ala in MAB_3543c and Thr39Ala in MAB_4408c). Of note, in the hybrid assembly obtained as a reference for all the analysis in Patient 14 (see Methods), two genes from the HNH endonuclease family were integrated in gene rearrangement events which are not present in the ATCC reference strain for *M. abscessus* ssp *abscessus*. In genotype 1 assembly, we have also detected two new genes from the HNH endonuclease family, known for their association with hypermutator phenotypes. These genes are implicated in reorganization events and notably absent in the ATCC reference.*

|

As the burst in diversity occurred since the end of the Genotype 1 stage, these findings might offer support to explain the burst of diversity identified in this patient.

We have included this new data and Discussion in the new version of the MS (lines 228-242 at the cleaned version or 251-266 at the changes control version). We have also detailed the 65 genes associated with hypermutation which have been used in this new analysis (Supplementary Table 4).

In addition, I recommend that the authors investigate the mutation spectra in this patient - as a skewed mutation spectra is strong evidence for hypermutation due to a defective repair system.

We appreciate the reviewer feedback regarding the mutation spectra in this patient, which allowed us to enrich our analysis. In fact, we have identified a biased mutation spectrum, which supports the likely involvement of a hypermutator phenotype, due to a defective repair system.

We analyzed the relative proportion of transitions and transversions in the 1168 substitutions (using as a reference the genotype 1 sequence) identified in the genotypes detected along the infection. A pronounced bias was detected in the mutation spectrum; notably, 1147 (98%) of these mutations correspond to transitions. Among these transitions, we found an enrichment in changes TA-CG (71%).

We have included these new relevant findings in Discussion (lines 353-365 at the cleaned version or 394-406 at the changes control version) to put these data in the context of hypermutation in Patient 14, and we have also added other references identifying similar skewed spectra associated to hypermutation in mycobacteria.

Reviewer #3 (Remarks to the Author):

Comments for the Authors

In the submitted manuscript, the authors use a comparative genomics approach to analyze sequential isolates of *Mycobacterium abscessus* (Mab) associated with 14 Patients (4 with Cystic Fibrosis). Although chronic infections (and sometimes re-infections) are common in this patient population, only a few studies have examined intra-patient diversity and microevolution of Mab. In the current study, the comparative genomics approach definitively identified at least one case of re-infection (Patient 13). However, the results suggest that, for the majority of infections, there were few changes between paired isolates, even after 2 months to several years of persistent infection. The major focus was on the case of Patient 14 who had 23 follow up isolates of *M. abscessus* subsp. *abscessus* over a 16 year period. The detailed analysis of those isolates highlights the complexity of long term Mab infections.

Major concerns include

1. Limitations of the bioinformatics methods
2. Limitations of the SNP analysis
3. Limitations of the plasmid analysis
4. Limitations in the presentation of the phylogenetic data

- **Bioinformatics Methods and SNP Analyses;**

The Methods should include additional details, such as the specific settings used with the various sequencing analysis tools.

All the Methods section has been thoroughly revised to include in the new version the specific settings used with each bioinformatic tool, as requested.

According to lines 358-369, a reference mapping approach was used. The Methods state that “Highly polymorphic and repetitive regions, phages, and PE/PPE regions were removed from the final SNP distance calculation and annotation” (Lines 336-337), but the Results don’t indicate what proportion of the reference genome was actually used or what proportion of the reads from each Patient isolate were discarded or leftover. The removal of those sequences, as well as the exclusion of “SNVs located in areas with a higher-than-expected number of calls”, suggests that the reported SNP distances are likely an underestimate of the absolute number of differences.

*In regards to the removal of highly polymorphic and repetitive regions, we cautionary eliminated them as they could potentially introduce biases in the analysis. In *M. tuberculosis*, these regions are systematically excluded. As for the “SNPs located in areas with a higher-than-expected number of calls”, this decision was taken to avoid regions with a tendency for false calls. We did not consider regions where, within a 10-nucleotide window, we detected more than 3 calls. These regions maybe more prone to accumulate sequencing errors which could lead to false calls. These errors often accumulate in challenging-to-sequence regions due to their composition or structure, or in repetitive or low-complexity regions, where read alignment can be problematic, potentially leading to erroneous calls.*

Once described the rationale behind the removal of these regions from the analysis, we, share the interest, raised by the reviewer, of indicating the proportion of genome impacted by this decision.

Regarding polymorphic and repetitive regions, a total of 46 genes encompassing repetitive regions (19), mobile elements or plasmids (9), and phages and prophages (18) were removed following genome annotation of the reference strain ATCC 19977. This removal accounted for 47,443 base pairs, constituting 0.97% of the genome.

Regarding the removal of areas with a higher-than-expected number of calls, it affected to a range between 1990 and 35240 bp, depending on the sequence. It corresponded to 0.04% to 0.69% of the total ATCC genome.

We believe that the loss of information resulting from the low percentage of genome not considered in the analysis, seems acceptable to be able to mitigate likely false positives. Filtering out these regions might mean just a minimum underestimation of the potential inpatient diversity. However, we have determined the specific SNPs removed in Patient 14 analysis due to the filtering of repetitive regions, PE/PPE regions and phages:

A total of 120 positions were affected. Among these, 115 represent mutations shared across all Patient 14 genotypes. The remaining five differential positions are as follows: 2 of them are shared by genotypes 9 and 12 both of which are in close relation as observed in the Bayesian analysis (Figure 5). One of these mutations mapped in a mobile element and the other in a gene coding for a phage endolysin gene. Another mutation is unique to genotype 12, located in the phage Gp37G968 family protein. The remaining two mutations are exclusive to genotype 7, one mapping in a mobile element and the other in the phage capsid and scaffold gene.

Regarding the specific SNPs not called due to the removal of windows with a higher-than-average number of calls, it only affected to 2 SNPs in patient 8 and another 2 in the Genotype 8 form Patient 14.

From these data it can be concluded that the impact of eliminating regions prone to false calls on the undetected diversity is really modest. We believe that this is a cost that can be assumed to assure proper precision in the SNPs that are called. We hope that the reviewer accepts our procedure once shown the minimum loss of information resulting for the elimination of uncertainty regions from our analysis.

The Methods indicate that SNP distances were calculated in comparison to the genome of the Mab Type strain ATCC 19977. However, mapping to a project-specific reference genome should provide a more accurate measure of SNP differences. Lines 389-391 indicate that hybrid assemblies were constructed for some isolates of Patient 14. Especially for the case of Patient 14, reference mapping to the hybrid assemblies derived from Genotype 1 and/or Genotype 2 datasets should allow more robust mapping of SNVs, in/dels and genomic rearrangement events, and improve characterization of the relationship between the numerous isolates and genotypes.

We fully agree with the reviewer that, due to Patient 14 genotypic complexity, a more precise variant calling should be performed, by using a project-specific reference genome instead of a reference ATCC strain. For this new version we have proceeded as suggested and a Genotype 1 assembly has been obtained (see new section in Methods defining the procedures). Although differences with respect to the number of calls obtained when using ATCC as a reference were quite similar, now we have recalculated all SNV calls for every patient 14 genotype, now using the Genotype 1 assembly as a reference (all new calculations have been included throughout the article; also, now in Supplementary table 2, two sheets for patient 14, using either the ATCC or the new assembly as references, can be found). The new Bayesian tree obtained for this patient (included in this new version, Figure 5) was also based in the usage of this assembly as a reference.

Regarding the potential rearrangements which could be detected when the new project-specific assembly has been used as the reference in the analysis of patient 14 genotypes, no rearrangements were identified during the intra-host evolution of genotypes. Regarding indels, five indels were detected, all deletions. Three of them were identified in node 1 (see Figure 5), one 10-base par deletion in node 2, branching into genotypes 3, 7, 9 and 12, and a final deletion in node 4.

Much of the Discussion is speculation about the relationships between the various Patient 14 genotypes. The presence of the 'fixed' A2270C mutation, the presence of two identical plasmids and the rough colony type associated with the 20 bp mps2 insertion across multiple genotypes is consistent with an infection evolving from a common ancestor since these findings (except for the plasmids) are rarely seen in first time unrelated isolates. The authors should determine if an erm (41) gene with the T28C substitution is present in the first, macrolide susceptible Genotype 1 strain as well as all the subsequent Patient 14 isolates of genotypes 1-12. In Bronson et al (ref 2), the Mab Type strain ATCC 19977 is assigned to Clade A1. In contrast, the T28C mutation is associated with a distinct set of circulating strains, Clade A3. The A2270C mutation accounts for 'reversion' to a macrolide resistant phenotype. However, the presence of T28C may provide clade information that determines the need to repeat SNV analyses with a more closely related (e.g., Clade A3) reference genome instead of ATCC 19977 (Clade A1)

Following the reviewer's request, we have now examined the presence of the T28C substitution and we can confirm that the wildtype allele is present in all the twelve patient 14 genotypes, ruling out that they correspond to Clade A3 and, therefore, validating the usage in our analysis of the ATCC 19977 strain as a closely related reference genome. Additionally, as we have now done a phylogenetic analysis of these genotypes (as a response of other reviewer's comments), we can also confirm that Patient 14 genotypes are clustered in the clade shared by the DCC1 sequences.

In general, the manuscript would benefit from additional details about the SNPs – including, but not limited to, the numbers of non-synonymous mutations, genes impacted by the nonsynonymous SNPs, and any association of the SNPs with specific genomic changes (in/dels, transposons, etc). Are the SNPs in 'core genes' or the accessory genome?

We have now detailed all the data requested by the reviewer in a new Supplementary table 2, which details whether the SNPs identified in all the 14 patients in study are i)

synonymous or non-synonymous, ii) the genes impacted by each SNP and iii) the protein the impacted genes codify for. Regarding the requested information of whether the SNPs are located in core genes or in the accessory genome, all them mapped in core genes. We believe that including also this last datum in the table might be reiterative; however, if the reviewer considers it necessary, we will proceed with its inclusion.

- **Plasmid Analysis**

The Abstract states that part of the purpose of the study was to characterize plasmids. It appears that plasmid analysis was only performed for isolates from Patient 14. If plasmid analysis wasn't performed for at least one isolate from every Patient, the authors should consider screening those datasets for sequences matching plasmid 1 (pJCM30620_1) and/or plasmid 2 (pS2c). Plasmids have only rarely been utilized as part of the genomic comparisons in mycobacteria and the additional plasmid analysis – as well as more discussion about these findings - would provide valuable information about the distribution of plasmids across Mab strains.

Following the insightful comment about plasmid characterization, we need to clarify the reasons behind having limited this analysis to only Patient 14. This is due because for this patient we had Nanopore long reads available, as a result of our interest to perform de novo assembly, to obtain a closer reference to an in-depth analysis of the diversity detected in this special case. As we had these long reads available, we could afford a more thorough characterization of plasmids, even closing them, something that could not be done for the remaining patients, as long reads were not available.

*As per the reviewer request, despite not being able to offer the same resolution in the analysis, we succeeded in tracking for the presence of plasmids 1 and 2. No plasmids were detected in the remaining cases except in the specimens from Patient 3, in whom we identified a single plasmid. By performing a BLAST analysis against the NCBI nt database, a match was found with *Mycobacterium abscessus* subsp. *abscessus* strain GD69A, plasmid pGD69A-1, with 99% coverage and 100% identity on the same contig. The Plasmids 1 and 2, mentioned by the reviewer were not found in any case.*

Now, we have included in the section Methods the approach followed by this analysis (lines 468-474 at the cleaned version or 534-540 at the changes control version) and in Results (lines 107-109 at cleaned version or 112-114 at the changes control version) the identification of a plasmid in patient 3. We must admit that we find it difficult to add additional Discussion around this limited finding.

In line 210, the authors refer to patient 4. This is likely a typo and should be 14.

Regarding the comment on line 210, indeed, it is a typographical error, and we intended to refer to patient 14, not patient 4. The error has been corrected in the manuscript.

- **Phylogenetic Analysis**

Figure 1 is a good depiction of the phylogeny of the sequenced isolates. However, it would be helpful to include lengths (# SNVs) for each branch as well as labels for each Patient 14 isolate (e.g., 14-1, 14-3).

We appreciate the reviewer's comment regarding the depiction of the phylogeny. Regarding the requested SNP-based lengths among patient 14 genotypes, these can be found in Supplementary Table 2. We can't include them now in Figure 1, as it is the requirement from the reviewer, because in the new version, in response to a comment from reviewer #2, we have conducted a new phylogenetic analysis, leading to a tree in which SNP-lengths can't be specified, but only phylogenetic distances (a scale of nucleotide divergence (SNP/site) has now been included to indicate distances). Following the other comment made by the reviewer, now, in each phylogeny, the different genotypes from the patients in study are indicated.

Figure 4 should also include branch lengths.

Now, also following a comment from another reviewer (Reviewer 1) we have substituted the network in Figure 4 by a Bayesian time tree (new Figure 5). This tree illustrates the relationship between genetic diversity and temporal signal, which causes a variation in the scale, that now has to be presented in years and means that, unfortunately SNV-lengths can't be indicated in the branches.

Consider combining the data from Tables 2 and 3 into a single table, or displaying that SNP and HZ data in Figure 4.

We have proceeded to merge the tables (Table 2) keeping the separation between the results corresponding to the genotypes and those for the intermediate nodes (using the tree in Figure 4 as a reference).

Do the various genotypes and phylogenetic clusters identified in the current study correlate with any of the clades described by Bronson et al (ref 2) Or sequence types from the Mab MLST scheme? (e.g., <https://pubmlst.org/organisms/mycobacteroides-abscessus-complex/>)

Thanks to the reviewer's comment, we have now included the STs (for the sequences in which it could be confidently assigned) in Table 1, along with the specification of whether the isolates belong to any clonal complex or dominant circulating clone (DCC). We have also now added these DCC data in Results (lines 171-176 and 319-326 at cleaned version, or 177-184 and 353-360 at the changes control version).

Regarding the clades described by Bronson, only five of our genotypes can be assigned to one of these groups, (all five belong to DCC A1). For the remaining genotypes, the distances between our genotypes and the references exceeded the number of SNPs indicated in the reference article to allow a precise assignation (Bronson, Ryan A., et al. "Global phylogenomic analyses of Mycobacterium abscessus provide context for non-cystic fibrosis infections and the evolution of antibiotic resistance." Nature communications 12.1 (2021): 514). Therefore, we believe that it is more informative to indicate only the DCC assignation in table 1, as mentioned before.

- **Additional comments**

Table 1: Remove the comment from Patient 14 that findings reflect "Mixed events (Persistence and Reinfection)" and change to "Persistence and Microevolution" since that is your conclusion presented as an "alternate conclusion" on page 13 of the Discussion.

We have proceeded with the correction suggested.

Do the hybrid assemblies include additional chromosomal sequences that didn't map to the ATCC 19977 reference genome? Although repetitive sequences and PE/PPE genes can compromise reference mapping, the presence/absence of specific prophage sequences could be phylogenetically informative.

We have now calculated the sequences in our hybrid assembly that don't map to the ATCC 19977 reference genome. While the reference genome has a total length of 5,067,172 base pairs, our assembly spans 5,521,121 base pairs, representing a difference of 453,949 base pairs (9% of the ATCC genome).

We then performed a qualitative analysis of this 9% of the genome, looking for specific prophage sequences, as requested by the reviewer. Only 19 phage-associated genes were identified, which, in our opinion, makes it difficult to use them with phylogenetical purposes.

- **Clinical Summary**

This paper is not about drug therapy of Mab and initial therapy was more than 15 years ago. However, since others will read about this therapy, some comments about its inadequacies need to be pointed out. Not to be overly critical, but only to be sure that a constructive and correct message is provided. Two of the drugs (TMP/SMX and fluoroquinolones) have no proven activity against Mab, whereas other drugs with proven activity (imipenem, linezolid, and tigecycline and now other better tolerated drugs omadacycline and eravacycline though the latter only recently available) were not reported as having been tried. In a species like Mab, which has only a single copy of the ribosomal rRNA operon, it's not unexpected that prolonged macrolide monotherapy resulted in the development of mutational resistance. With macrolide susceptible isolates, we now know that appropriate treatment involves a macrolide plus a proven (IV) companion drugs and that prolonged combination therapy comes with a high expectation of cure. The patient did receive cefoxitin and amikacin, but only a month at a time, which is clearly an inadequate duration. The Discussion should include additional comments about therapy, especially about avoidance of macrolide monotherapy and (for infections with macrolide susceptible Mab) promotion of combination therapy with macrolides plus proven companion (IV) meds.

We fully agree with the reviewer's comments. Unfortunately, most of the initial treatment decisions were made years ago by clinicians others than those currently managing this patient and that lied beyond our control.

We must also remember that Patient 14 is a CF case, which means really long-term treatments (more than 16 years in this case) and some of the therapeutical decisions are consequence of i) the fact that the patient was a child when starting treatment, ii) the coexistence of other coinfections and iii) the need to assure adherence and tolerance in a long-term scope. That meant that several constraints had to be considered when deciding the most suitable therapeutical decisions.

With regard to macrolides, the medical team discussed this extensively at the time, and the decision was unanimous, as it was the only way to maintain the patient's clinical stability. Ultimately, this is an infection caused by M abscessus in the context of cystic fibrosis,

a condition that behaves in a characteristic manner, and at times, deviating from established guidelines is necessary to keep patients stable. In fact, thanks to this approach, the patient remained stable for many years without requiring additional reinforced treatments.

Regarding the use of levofloxacin, while it may not be the optimal therapy, there are some references suggesting its potential use (Varley C, Winthrop K. Nontuberculous Mycobacteria: Diagnosis and Therapy. Clin Chest Med. 2022;43(1):89-98). Additionally, there are treatment guidelines for M. abscessus infections in cystic fibrosis that include the use of moxifloxacin (Andrew E, Connell T, Robinson P, et al. Mycobacterium abscessus in cystic fibrosis. Journal of Paediatrics and Child Health 55 (2019) 502–511). Currently, inhaled levofloxacin is being used for M. abscessus treatment in patients with cystic fibrosis who have poor tolerance to inhaled amikacin.

As for trimethoprim-sulfamethoxazole (TMP-SMX), again, given the available options and the patient's tolerance, it was considered the best choice.

Finally, regarding the 4-week treatment duration, at that time, the patient was a child, and the decision to limit the duration of the induction treatment was made to mitigate potential toxicities due to her age and tolerance.

We have now included a new paragraph in Discussion (lines 264-283 at clean version or 297-316 at the changes control version) summarizing (more extensively than here) all these facts and also adding the comments and requests raised by the reviewer.

REVIEWERS' COMMENTS

Reviewer #1 (Remarks to the Author):

No further comments, everything has been addressed.

Reviewer #2 (Remarks to the Author):

Thank you to the authors for making changes to this manuscript. I am glad that the investigation into hypermutation has enhanced this work. However I do think more improvement is required:

line 8 - "Not too many" should be "Few"

line 194 - isolates should be from a single colony, and these have mixed alleles therefore the term isolate is not appropriate here

line 108 onwards - very confusing use of the term "Genotype" - how is genotype defined? This is not stated anywhere. In figure 5 it appears if each genome is defined as a different genotype (even if only 1 snp different). This is not the usual use of genotype as some SNP diversity would be expected in all chronic infections and yet would be considered the same "genotype". I suggest the authors find another way of describing the SNP diversity within patient 14

line 214 - please provide the locus IDs (MAB_XXXX) for the two HNH endonuclease family genes

Reviewer #3 (Remarks to the Author):

In this study, the authors used a genomics approach to examine sequential isolates of *Mycobacterium abscessus* from 14 patients with respiratory infections. In twelve cases, minimal genomic diversity (0-12 differential SNVs) was observed, suggesting that most illness was due to persistent infections. In one case, extensive variation (15,956 SNVs) was observed, which indicates that re-infection can occur. The final case, which involved 23 sequential isolates collected over 192 months, revealed a complex pattern of microevolution.

The current version of this manuscript includes extensive revisions to the methods, text and figures. These changes improved the quality of the data analysis and robustness of the conclusions.

Reviewer #4 (Remarks to the Author):

Answers to Reviewer

line 8 - "Not too many" should be "Few"

The change has been made, as suggested (line 35, non-blind version)

line 194 - isolates should be from a single colony, and these have mixed alleles therefore the term isolate is not appropriate here

We have now modified this sentence, following the reviewer suggestion to: "...six sequential cultures in which we had identified genotype 2", to avoid the use of the term "isolate" (line 187, non-blind version)

line 108 onwards - very confusing use of the term "Genotype" - how is genotype defined? This is not stated anywhere. In figure 5 it appears if each genome is defined as a different genotype (even if only 1 snp different). This is not the usual use of genotype as some SNP diversity would be expected in all chronic infections and yet would be considered the same "genotype". I suggest the authors find another way of describing the SNP diversity within patient 14

*Following the reviewer's comments we have now defined in Methods (line 443, non-blind version) that we use the term 'Genotype' when the number of SNPs genomic exceeded the genomic threshold (25-30 SNPs as we also specify and reference in line 260 in Discussion) applied in the literature to consider two *M. abscessus* isolates as different strains. Therefore, we have never applied the term "genotype" for shorter distances. We must clarify that the assignation of genotypes in Figure 5 also followed this same criterion.*

We hope to have supported the proper usage of the term "genotype" after specifying the threshold in the number of SNPs used to define different genotypes. However, if the reviewer still have concerns regarding this issue, please let us know and we will try to describe diversity within patient 14 in a different way.

line 214 - please provide the locus IDs (MAB_XXXX) for the two HNH endonuclease family genes

*We have incorporated the Protein-Encoding Genes IDs associated with the assembly annotation and their best identities in the NCBI's nr database (lines 239). Since the assembly differs from the *Mycobacterium abscessus* ATCC strain, we have chosen to maintain the locus identification extracted from the annotation instead of the MAB ID. However, upon reviewer's preference, we are open to substituting the PEG ID with the recommended MAB ID from the assembly.*